# High Mountain Asian glacier response to climate revealed by multi-temporal satellite observations since the 1960s

Atanu Bhattacharya [1,2 ✉], Tobias Bolch [1 ✉], Kriti Mukherjee [3], Owen King[1], Brian Menounos[3], Vassiliy Kapitsa[4], Niklas Neckel[5], Wei Yang[6] & Tandong Yao [6]

Knowledge about the long-term response of High Mountain Asian glaciers to climatic variations is paramount because of their important role in sustaining Asian river flow. Here, a satellite-based time series of glacier mass balance for seven climatically different regions across High Mountain Asia since the 1960s shows that glacier mass loss rates have persistently increased at most sites. Regional glacier mass budgets ranged from $-0.40 \pm 0.07$ m w.e.a$^{-1}$ in Central and Northern Tien Shan to $-0.06 \pm 0.07$ m w.e.a$^{-1}$ in Eastern Pamir, with considerable temporal and spatial variability. Highest rates of mass loss occurred in Central Himalaya and Northern Tien Shan after 2015 and even in regions where glaciers were previously in balance with climate, such as Eastern Pamir, mass losses prevailed in recent years. An increase in summer temperature explains the long-term trend in mass loss and now appears to drive mass loss even in regions formerly sensitive to both temperature and precipitation.

[1] School of Geography and Sustainable Development, University of St Andrews, Scotland, UK. [2] Department of Remote Sensing & GIS, JIS University, Kolkata, India. [3] Geography, Earth and Environmental Sciences, University of Northern British Columbia, Prince George, Canada. [4] Institute of Geography, Ministry of Education and Science, Almaty, Kazakhstan. [5] Alfred-Wegener-Institut Helmholtz-Zentrum für Polar und Meeresforschung, Bremerhaven, Germany. [6] Institute of Tibetan Plateau Research, Chinese Academy of Sciences, Beijing, China. ✉email: atanudeq@gmail.com; tobias.bolch@st-andrews.ac.uk

High Mountain Asia (HMA) contains the largest concentration of glaciers and ice caps outside the Polar regions and is considered the water tower of Asia[1] with major Asian rivers relying on the continued supply of meltwater from these glaciers[2–4]. This water tower is, however, vulnerable to the effects of climate change and projected increases in temperature will drive glacier recession until the end of this century[5,6]. After an initial increase in runoff due to glacier wastage, continuous glacier retreat and subsequent disappearance will lead to a significant decrease in runoff in most regions[5–8] in the coming decades. Yet, projections of future glacier area and volume, and their runoff yield, come with high uncertainties, as numerous gaps remain in our knowledge of regional glacier behavior and the interaction of glaciers with a changing climate[5,9,10]. To improve our understanding of HMA glacier response to climate, detailed and robust long-term observations are needed. Long-term mass balance data are particularly valuable as they are used to calibrate and validate glacier mass balance models. The traditional way of estimating glacier mass balance is through in situ observations, also known as the direct glaciological method. However, in situ measurements exist for only about 30 out of more than 80,000 individual glaciers in HMA[5]. A limited number have observations of 5 years or more and only two glaciers have mass balance records that exceed 30 years. Both of these glaciers are located in the Tien Shan (Tuyuksu and Urumchi No. 1 glaciers), and there is none in the remaining vast area of HMA[5,11]. An alternative method to calculate glacier mass changes over large regions is the measurement of glacier surface elevation changes from space ("geodetic glacier mass balance"). Satellite laser altimetry measurements and the processing of full stereo satellite imagery archives strongly enhanced our understanding of glacier mass changes in HMA since 2000 (e.g. ref. [12]. vs. ref. [13]. and refs. [5,14,15]. vs. refs. [16,17]). These measurements show that, in line with the global pattern of glacier response to climate change[18], most glaciers in HMA lost mass over the last two decades. Glaciers in the Central Karakoram, Western Kunlun, and Eastern Pamir region, however, remained in balance until recently, a phenomenon which is often referred to as the 'Karakoram anomaly'[19]. Valuable insights into glacier behavior since the mid-1970s have been made using US spy satellite Hexagon KH-9 imagery[20–26]. These studies confirmed that, in general, similar spatial heterogeneity of glacier mass changes existed before 2000, but also that an increase in mass loss rates occurred in most regions after 2000. However, these studies investigated a maximum of two extended (up to thirty year) periods only, and thus the link between glacier fluctuations and the regions' changing climate remains loosely constrained. These previous studies show the benefits of the use of Hexagon KH-9 imagery in understanding broad-scale glacier change. Poor contrast and low dynamic range, however, hamper their accuracy[20,27] and leads to errors especially in the higher reaches of many glaciers where large data voids of >40% of glacier area are not uncommon. Declassified stereo imagery acquired by the Corona KH-4 satellite in the 1960s and early 1970s[28,29] is of high (~2 m) spatial resolution, which increases the probability of image matching during DEM generation. Moreover, this data provides the opportunity to capture glacier elevation change during a period when no other high resolution stereo data were available[30,31]. Although Corona KH-4 data were first employed to study glacier elevation changes over a decade ago in the Mt. Everest region[32,33], and more recently in a few localized studies[34–37], much of this archive remains untouched, primarily due to the complex geometry and high level of distortion (Supplementary Fig. 1) present in the imagery.

Here we analyse multi-temporal geodetic glacier mass budgets in seven climatically different glacierised regions over a timespan of almost six decades to characterize glacier mass budget variability across HMA (Fig. 1). These geodetic mass balance estimates have been derived from DEMs generated from available satellite data including Corona KH-4, Hexagon KH-9 and more recent spaceborne data including high-resolution Pléiades imagery (Supplementary Note 1 and Supplementary Table 1). Additionally, we examine influential climate drivers in different regions using ERA5 Land and in-situ weather station data to improve our understanding of the response of HMA glaciers to a changing climate. Our results reveal accelerated mass loss over the seven study areas, even in the regions with previously balanced mass budgets, due to widespread increases in summer temperature.

## Results and discussions

**Regional glacier changes.** Substantial glacier mass loss occurred in each of the seven regions (Figs. 1, 2, Supplement Figs. 2–17), although much temporal variability is exists in the evolution of glaciers towards their current state. In some regions elevated and sustained glacier mass loss is evident throughout the satellite era; glaciers in the Ak-Shirak range/Central Tien Shan lost mass over the whole investigated period (1964–2019) at a maximum rate of $-0.54 \pm 0.10$ m water equivalent per annum (w.e.a$^{-1}$) between 1980 and 2002. Several areas experienced consistently increasing mass loss rates from our earliest mass loss observations to the present day (Fig. 2, Supplementary Tables 2–9). In the Northern Tien Shan, Western Nyainqentanglha (Central-East Tibetan Plateau) and the Poiqu region including the Langtang catchment (Central Himalaya) moderate mass loss rates ($-0.18 \pm 0.11$ to $-0.30 \pm 0.10$ m w.e.a$^{-1}$) occurred over the era of Corona KH-4 and Hexagon KH-9 imagery (1960s–1980s). Mass loss rates more than doubled between the earliest ($-0.18 \pm 0.11$ m w.e.a$^{-1}$ for 1964–1971) and most contemporary (2016–2020) time periods in Northern Tien Shan ($-0.49 \pm 0.13$ m w.e.a$^{-1}$) and increased also markedly in Western Nyainqentanglha ($-0.46 \pm 0.14$ m w.e.a$^{-1}$ for 2012–2019), Poiqu ($-0.42 \pm 0.11$ m w.e.a$^{-1}$ for 2004–2018) and Langtang sub-region ($-0.58 \pm 0.11$ m w.e.a$^{-1}$ for 2015–2019). Similar to the whole Poiqu region, glacier mass loss rates increased in the Langtang sub-region when compared to earlier time periods (Supplementary Table 7). We calculate a mean mass budget of $-0.23 \pm 0.10$ m w.e.a$^{-1}$ from 1964 to 2004, which has probably doubled ($-0.50 \pm 0.11$ m w.e.a$^{-1}$) by the most recent period (2004–2019).

In some regions, recent glacier mass budgets transitioned from mass gain to mass loss. At Gurla Mandhata/West-Central Himalaya, for example, we measured only slight mass loss from 1966 to 2011 ($-0.12 \pm 0.10$ m w.e.a$^{-1}$) and found balanced budgets from 2011 to 2013 ($-0.02 \pm 0.08$ m w.e.a$^{-1}$). Following 2013, however, rates of mass loss from these glaciers increased to $-0.20 \pm 0.11$ m w.e.a$^{-1}$ (2013–2016) and $-0.24 \pm 0.17$ m w.e.a$^{-1}$ in our most recent time period (2018–2019). Glaciers draining the Purogangri ice cap (Central Tibetan Plateau) also displayed a period of almost balanced mass budgets, albeit over a slightly earlier period (2000–2012), but mass loss was observed before and after this period. Slight glacier mass loss occurred from glaciers in Muztagh Ata Massif/Eastern Pamir between 1967 and 1973, but balanced and even slightly positive glacier mass budgets prevailed between 1973 and 2009. Thereafter, however, the glaciers lost mass at an increasing rate ($-0.12 \pm 0.09$ m w.e.a$^{-1}$ from 2013 to 2019) again indicating the transition from a quasi-balanced glacier mass budget to glacier mass loss in this region towards the present day (Fig. 2). Our results closely match those of previous studies for overlapping time periods (see Fig. 3 and section "Comparison with other mass balance estimates"). However, our analyses extend these records back in time and towards the present day and also add intermediate periods.

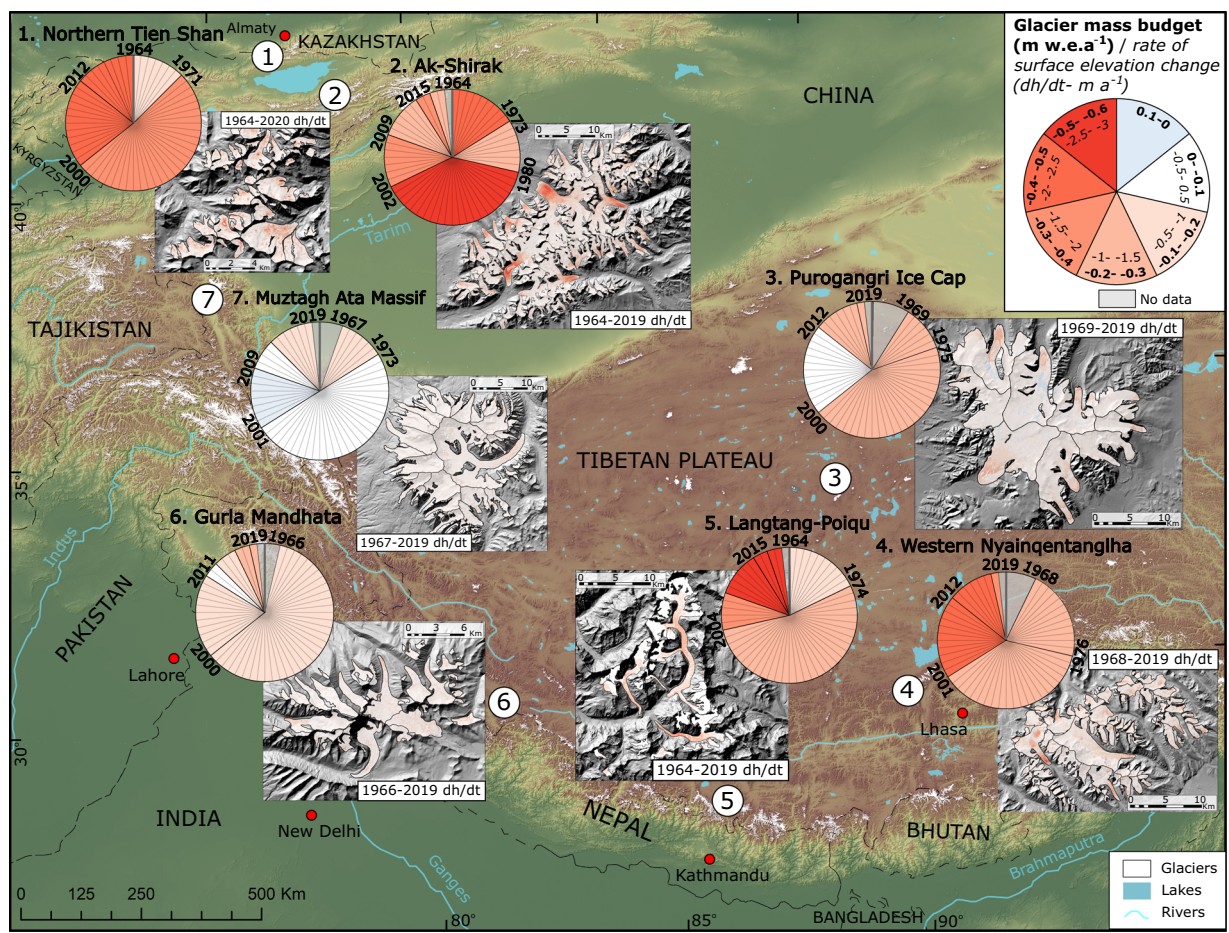

**Fig. 1 Location of the seven study regions across High Mountain Asia, glacier mass budget estimates for all available different periods and associated glacier surface elevation change rates (for each pixel and the longest available period) for each study site.** Vertical gray lines mark the start date (1964) of the study period on pie charts. High mass loss is especially evident in Tien Shan (regions 1 and 2) and Central Himalaya (region 5). Especially the Eastern Pamir (region 7), but also Central Tibetan Plateau (region 3) experienced longer periods of near zero mass changes which are now over (*Figure prepared by Owen King*).

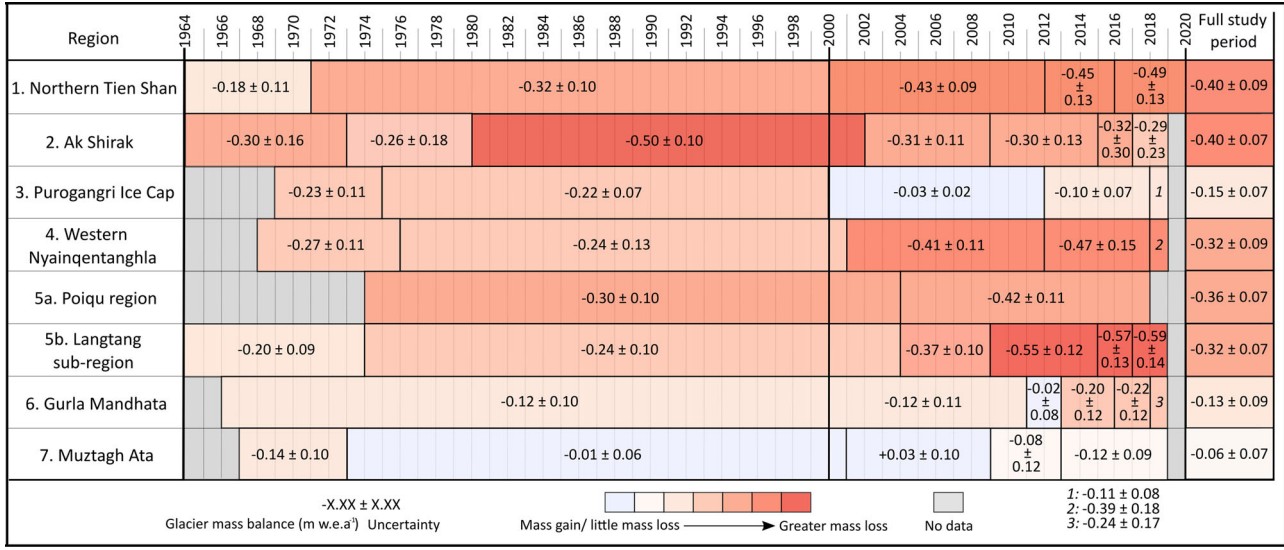

**Fig. 2 Summary of the geodetic glacier mass budgets in different time periods for all the studied regions along High Mountain Asia (HMA).** Notwithstanding temporal heterogeneity over the entire study period, extensive ice-mass loss has occurred in all the seven study regions. Several regions, such as, Northern Tien Shan, Western Nyainqentanghla and Poiqu/Langtang have experienced consistent increase in mass loss rates throughout the observation periods. In Muztag Ata, where glaciers had lost little mass for several decades, glacier mass loss has occurred in the recent time span.

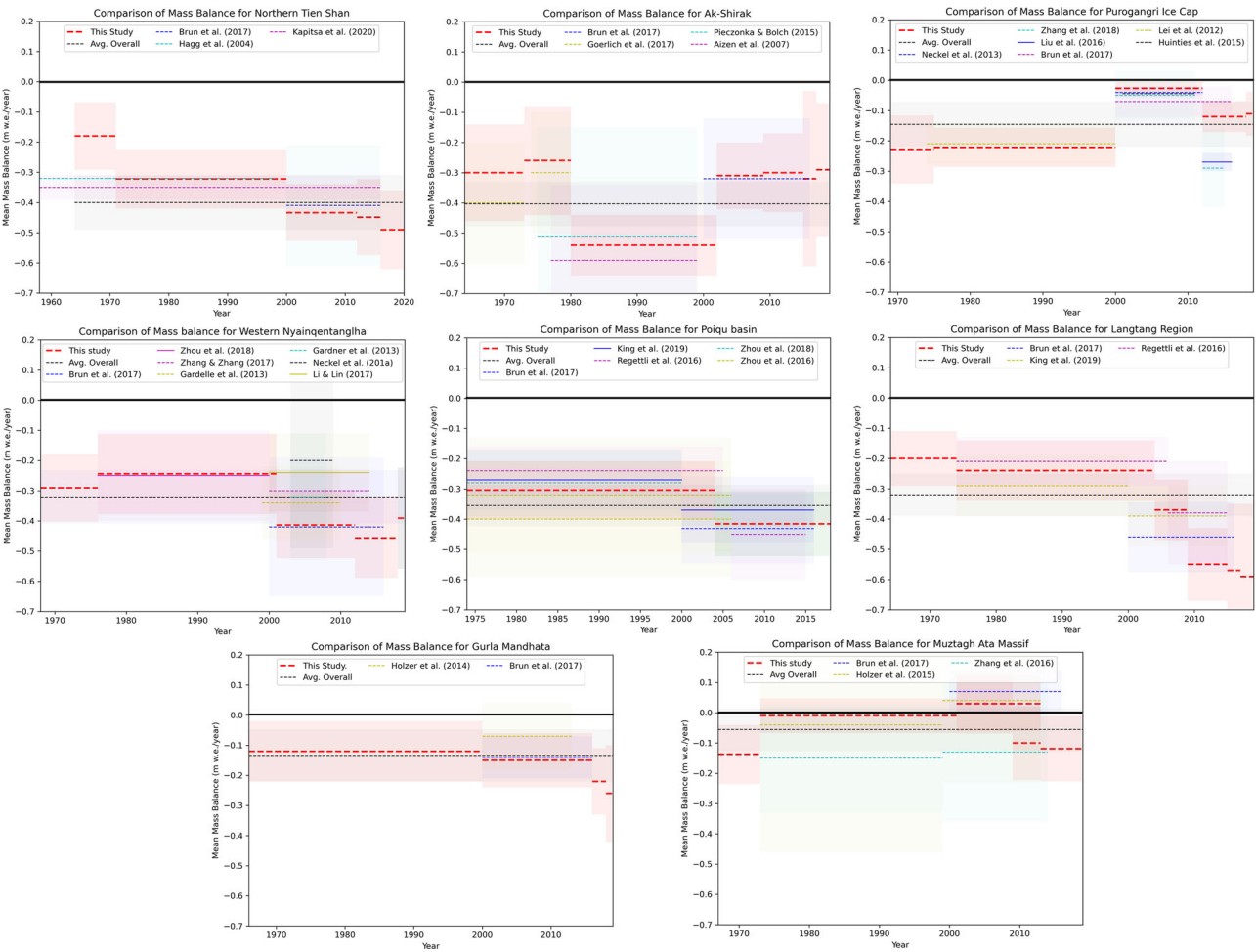

**Fig. 3 Comparison of regional glacier mass balance with other published studies.** Figure 3 indicates mean glacier mass balance (dotted lines) and associated uncertainty (light color rectangle). Considering the temporal and methodological variability, the regional mass balance pattern for the present study is in general consistent with the other available studies. Balanced mass budget periods appear to have ended in Muztag Ata, Gurla Mandhata, and Purogangri Ice Cap regions and the glaciers of these regions have lost mass recently.

Glacier area changes followed the patterns of the overall mass changes in all regions (Supplementary Tables 10–16). The highest area loss occurred in the Northern Tien Shan ($-0.60 \pm 0.07\%\,\mathrm{a}^{-1}$) and Western Nyainqentanglha ($-0.30 \pm 0.02\%\,\mathrm{a}^{-1}$). No significant area changes occurred alongside the close-to-balanced mass budgets found in the Gurla Mandhata ($-0.08 \pm 0.03\,\mathrm{km}^2\,\mathrm{a}^{-1}$) and Muztagh Ata Massif ($-0.07 \pm 0.02\,\mathrm{km}^2\,\mathrm{a}^{-1}$) regions. Glacier shrinkage for other study areas, such as, Ak-Shirak ($-0.19 \pm 0.01\%\,\mathrm{a}^{-1}$), Purogangri ($-0.17 \pm 0.01\%\,\mathrm{a}^{-1}$) and Poiqu ($-0.18 \pm 0.01\%\,\mathrm{a}^{-1}$) were within a similar range.

**Regional glacier response to climate change.** To better understand how climate and climate change drives HMA glacier mass balance we characterised the general climate in the seven study regions and compared our records of mass changes to surface fluxes from a regional model (ERA5 Land) forced with reanalysis data[38]. The regions Northern Tien Shan, Ak-Shirak (Central Tien Shan) and Muztagh Ata (Eastern Pamir) are mainly influenced by mid-latitude westerlies. While the former is more humid the latter two regions are relatively dry (Supplementary Note 2). The regions Purogangri Ice Cap and Western Nyainqentanglha on the Tibetan Plateau are located in the drier transitional zone between the mid-latitude westerlies and monsoon influenced regions. The Poiqu/Langtang and Gurla Mandhata regions are located in the humid Central Himalaya, where both the Indian and South-East Asian summer monsoon govern meteorological conditions.

We extracted temperature and precipitation estimates from the ERA5 Land grid cells representing glacierised elevations for each sub region of our study (Supplementary Table 17) and specifically considered summer air temperature (SumT) and solid precipitation (snow fall, SolP: considering any precipitation corresponding to temperature <0 °C as snow) in our analyses since these two meteorological variables are the most important drivers of glacier mass balance (Supplement). Increased SumT anomalies (deviation from the mean over the whole study period in each region) are particularly pronounced and are therefore likely to be the main driver of enhanced ice loss[39,40], although subtle decreases in SolP may have also enhanced ice loss rates in some regions (Fig. 4, Supplementary Table 17, Supplementary Figs. 21–24).

To validate ERA5 Land data and to obtain climatic information for the pre-ERA5 Land period we compiled in-situ measurements from proximal weather stations with data availability since at least the 1970s (Supplementary Table 18). Temperature trends and anomalies between the two datasets show good agreement ($r^2$ ranges from 0.39 to 0.91 and $p$ ranges from 0.05 to 0.001) for most of the time series (Supplementary Figs. 18–20). Precipitation showed little agreement but we did find high correlation for winter precipitation of Gurla Mandhata ($r^2 = 0.82$ and $p = 0.01$).

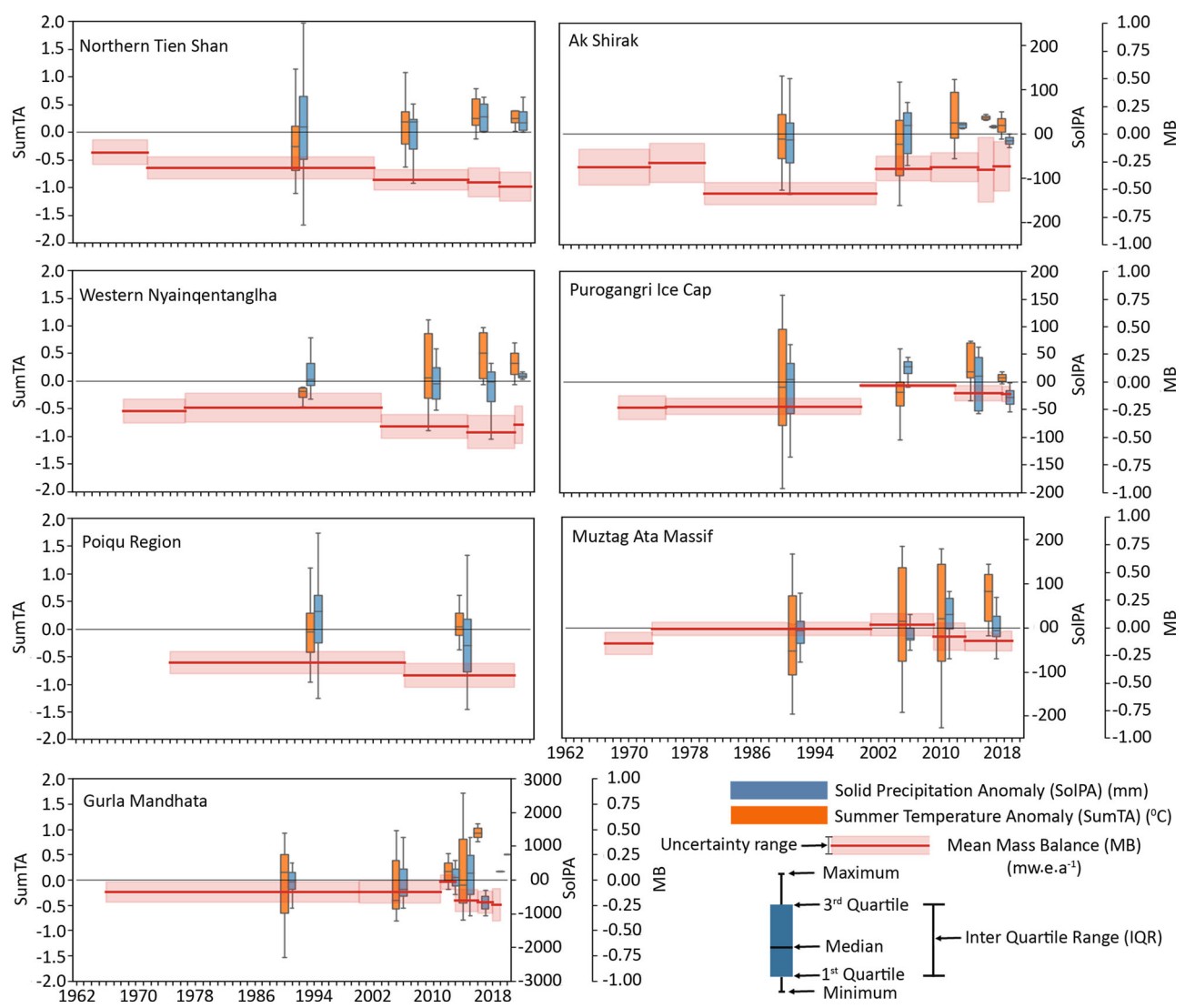

**Fig. 4 Comparison of average annual mass change rate of glaciers with ERA5 Land climate variables for all study regions in different time periods.**
Summer temperature and solid precipitation anomalies of ERA5 Land gridded climate data (available since 1981) along with geodetic glacier mass balance estimates for the different regions (left column: more humid regions; right column: drier regions) and each time period.

The generally weaker agreement of precipitation records (Table 1) likely reflects the relatively sparse network of weather stations and their typical location, aside from Tuyuksu (Northern Tien Shan), at lower elevations than glacierised terrain. Thus, in-situ measurements may not be fully representative of precipitation received by glaciers. In combination with the well documented difficulties associated with recording precipitation, high elevation meteorological stations may be more biased than those at lower elevations[41,42]. On the other hand, the ERA5 Land reanalysis product is obtained by combining model data with observations considering the laws of physics to produce a globally consistent dataset[43]. To allow for the comparison of our mass balance data against a spatially and methodologically homogenous record of meteorological variables, we conduct further analyses considering the ERA5 Land dataset primarily, but also provide a comparison of our results to meteorological station records (Table 1).

The correlation between glacier mass balance estimates and meteorological variables derived from ERA5 Land reanalysis data confirmed SumT to be the primary control on glacier mass budgets in Northern Tien Shan ($r^2 = 0.97$ and $p = 0.009$). A similarly strong correlation between $SumT_{in-situ}$ and glacier mass budgets exists for the measurements taken at the Tuyuksu

meteorological station ($r^2 = 0.92$, $p = 0.004$) (Table 1). The relationship between SumT and glacier mass loss is clearly evident from 2000 to 2012 when a strong SumT increase forced a reduction in SolP, as most precipitation occurs in late spring and summer in this region. No single climatic variable had a significant impact on glacier mass budgets when other regions were considered individually. We, therefore, grouped the remaining regions in two clusters based on their general climatic characteristics and estimated the correlation between these compiled samples of glacier mass balance and climatic variables. Annual precipitation and temperature data from ERA5 Land show that the mainly monsoon influenced Himalayan regions Poiqu and Gurla Mandhata have a humid climate. In these regions summer is the main precipitation season and the mean annual precipitation ranges from 1710 mm a$^{-1}$ to 5220 mm a$^{-1}$. Mean annual temperatures (MAAT) are between −2.4 °C and −7.9 °C. We also included Western Nyainqentanglha in this grouping (860 mm a$^{-1}$ and −3.75 °C) with summer also being the main precipitation season. Ak-Shirak, Purogangri Ice Cap, and Muztagh Ata Massif are typified by a cold, dry climate due to lower annual rates of precipitation (415 mm a$^{-1}$ to 607 mm a$^{-1}$) and low annual temperature (−8.8 °C to −9.7 °C).

**Table 1 Correlation between different meteorological variables extracted from weather station observations, ERA5 Land and glacier mass balance estimates for different regions and climatic regimes.**

| Sl. No | Parameters | Northern Tien Shan (Tuyuksu station) | | Humid climatic regions (GM, Poiqu, WN) | | Humid climatic regions (GM, Poiqu) | | Cold & Dry climatic regions (AK, PG, MA) | |
|---|---|---|---|---|---|---|---|---|---|
| | | $R^2$ | $p$ | $R^2$ | $p$ | $R^2$ | $p$ | $R^2$ | $p$ |
| Weather station data | | | | | | | | | |
| 1 | Summer temp | **0.92** | **0.004** | **0.53** | **0.002** | **0.77** | **0.004** | **0.60** | **0.00005** |
| 2 | Summer Precipitation | 0.67 | 0.06 | 0.52 | 0.15 | 0.57 | 0.23 | 0.39 | 0.64 |
| 3 | Winter Precipitation | 0.42 | 0.15 | **0.51** | **0.004** | **0.58** | **0.03** | 0.41 | 0.17 |
| 4 | Annual Precipitation | 0.59 | 0.06 | 0.38 | 0.15 | 0.62 | 0.27 | 0.44 | 0.83 |
| ERA5 Land Reanalysis grided climate data | | | | | | | | | |
| 1 | Summer temp | **0.97** | **0.009** | **0.71** | **0.003** | **0.69** | **0.001** | 0.16 | 0.10 |
| 2 | Solid Precipitation | 0.12 | 0.85 | **0.64** | **0.0001** | **0.56** | **0.001** | **0.79** | **0.0007** |
| 3 | Summer Precipitation | 0.01 | 0.86 | 0.66 | 0.3 | 0.60 | 0.2 | 0.19 | 0.15 |
| 4 | Winter Precipitation | 0.62 | 0.42 | **0.66** | **0.003** | **0.61** | **0.001** | **0.53** | **0.004** |
| 5 | Annual Precipitation | 0.19 | 0.78 | **0.64** | **0.003** | **0.56** | **0.001** | **0.62** | **0.02** |

Bold numbers indicate strong correlation between glacier mass balance and meteorological variables.
*GM* Gurla Mandhata, *WN* Western Nyainqentanghla, *AK* Ak-Shirak, *PG* Purogangri, *MA* Muztagh Ata.

In cold, dry regions, glacier mass budgets were most strongly influenced by SolP. We find a particularly high correlation ($r^2 = 0.79$, $p = 0.0007$) between SolP in ERA5 Land data and glacier mass balance in such regions (Table 1). However, the influence of temperature change could also be important here, as summer temperatures measured at weather stations showing substantial ($r^2 = 0.60$, $p = 0.00005$) correlation with glacier mass balance estimates. The almost balanced mass budget of glaciers of the Purogangri Ice Cap from 2000 to 2012 suggests that changes in SolP affect decadal-scale variations in mass balance. This period of reduced ice loss coincides with an increase in SolP (42 mm a$^{-1}$, 15%) and a slight decrease in SumT ($-0.24\,°$C, Supplementary Table 17) occurred over the same period. At Muztagh Ata Massif, one of the coldest and driest regions in HMA, periods of ice mass loss have coincided with increases in SumT for two periods. SumT increase between the 1960s and 1970s ($\sim0.39\,°$C) and after 2009 (Supplementary Fig. 20 and Supplementary Table 17) accompanied increased rates of glacier mass loss. An increase in SumT ($\sim0.21\,°$C) and decreased SolP ($\sim10$ mm a$^{-1}$) have likely driven mass loss after 2013 (Supplementary Table 17). This region also experienced a slightly positive mass balance ($+0.03 \pm 0.10$ m w.e.a$^{-1}$) during 2001–2009 when an only insignificant increase in SumT ($\sim0.04\,°$C) occurred. This period of slight mass gain was probably the effect of increased SolP (42 mm a$^{-1}$).

In the more humid climatic regions (Poiqu, Gurla Mandhata and Western Nyainqentanglha), glacier mass budgets appear to be equally influenced by summer temperatures (Table 1) (ERA5 Land: $r^2 = 0.71$, $p = 0.003$; Weather station: $r^2 = 0.53$, $p = 0.002$) and SolP (ERA5 Land: $r^2 = 0.64$ and $p = 0.0001$), annual (ERA5 Land: $r^2 = 0.64$ and $p = 0.003$) and winter (ERA5 Land: $r^2 = 0.66$, $p = 0.003$ and Weather station: $r^2 = 0.51$, $p = 0.004$) precipitation. Glaciers in Gurla Mandhata, which receive substantial amounts ($\sim1900$ mm a$^{-1}$) of SolP from the Indian summer Monsoon and winter Westerlies displayed approximately balanced budgets from 2011 to 2013, but a substantial perturbation in SumT since 2013 ($\sim0.8\,°$C during 2013–2018) appears to have destabilised glacier mass budgets here (Supplementary Table 17). Glaciers in the regions where mass loss has occurred after a period of balanced glacier mass budgets, such as Gurla Mandhata and Muztagh Ata Massif, seem highly sensitive to marked shifts in SumT, which overrides the longer-term influence of variables such as SolP.

The transition towards substantial mass loss from glaciers in Western Nyainqentanglha from 2001 to 2012 (Fig. 4) occurred in response to concomitant increases in SumT ($\sim0.23\,°$C) and decreases in SolP ($\sim25$ mm a$^{-1}$) (Fig. 4), suggesting similar dual sensitivity of glaciers here to temperature and precipitation variability as in Gurla Mandhata or the Poiqu basin. However, the location of Western Nyainqentanglha in a transition zone where both the Indian summer monsoon and mid-latitude westerlies exert an influence on local climate likely complicates the relationship between glacier mass balance and climate variability here[44,45]. The correlation between glacier mass balance and both annual or summer temperature (ERA5 Land: $r^2 = 0.68$–$0.69$, $p = 0.001$ in both cases) and annual ($r^2 = 0.56$, $p = 0.001$) or winter precipitation ($r^2 = 0.61$, $p = 0.001$) remains substantial in other humid regions (Gurla Mandhata and Poiqu) when Western Nyainqentaglha is removed from these analyses. This suggests no stronger correlation between glacier mass balance and annual or summer temperature, or between glacier mass balance and annual and winter precipitation, in western Nyainqentaghla when compared with Gurla Mandhata or Poiqu basin. Additional factors, such as the variable influence of the mid-latitude westerlies[44] or Indian summer monsoon onset timing[46], could also have significant influence on glacier mass loss variability in western Nyainqentanglha, which have not been exclusively studied here.

When considered collectively, it appears that recent, pervasive increases in SumT (Fig. 4) have driven both the acceleration of ice mass loss in several regions across HMA, as well as the transition from periods with close to zero mass balance to mass loss in other regions (Muztagh Ata Massif, Purogangri ice Cap, Gurla Mandhata). Our most contemporary geodetic mass balance estimates show that glacier mass loss is occurring in all seven of our study sites across HMA in consequence.

**Comparison with other mass balance estimates.** Regional scale geodetic studies[14–16,47] established the heterogeneous pattern of glacier response across HMA due to contrasting climate settings. Our contemporary (2000–2019) estimates of mass balance are similar to those of Brun et al.[16] and Shean et al.[17]; our geodetic estimates differ from these earlier studies by only ±0.02 and ±0.04 m w.e.a$^{-1}$, respectively. Our mass balance estimates are also in general consistent with other more regionally focused studies and in-situ measurements over comparable time periods (Fig. 3) as well as with those of different methodologies or data sources (for detailed comparisons see Supplementary Note 3, Supplementary Figs. 25–32 and Supplementary Tables 20–27) and offer a temporally resolved view of the evolution of glaciers towards their current state across HMA.

In Northern Tien Shan, strong mass loss had previously been highlighted[16] ($-0.41 \pm 0.20$ m w.e.a$^{-1}$) until 2016, and our recent

data suggested increasing rates of mass loss ($-0.49 \pm 0.13$ m w.e.a$^{-1}$) have continued over our most recent period (2016–2020). Specific mass balance estimates for Tuyuksu Glacier, Northern Tien Shan, which has been closely monitored since 1957 using the glaciological method, vary slightly depending on the methods used[16,48]. Our geodetic measurements, along with those of other studies[48,49], suggest less negative mass balance when compared to in-situ observation (Supplementary Fig. 25), which indicates a possible need to reanalyze the glaciological measurements[48].

Glaciers in Ak-Shirak experienced substantial mass loss over the entire study period ($-0.40 \pm 0.07$ m w.e.a$^{-1}$). Our data suggest maximum ice mass loss rates ($-0.54 \pm 0.10$ m w.e.a$^{-1}$) between 1980 and 2002, akin to other geodetic observations covering similar periods[50,51] (Fig. 3). This period of enhanced ice loss may be due to heightened glacier surface lowering following several glacier surges before 1980 (Supplementary Table 17 and next section).

The mass loss rate of glaciers draining the Purogangri Ice Cap fluctuated at an almost decadal timescale, although moderate mass loss ($-0.15 \pm 0.07$ m w.e.a$^{-1}$) prevailed over the entire observation period (1969–2019). Our estimated ice loss rates for the period 1975–2000 ($-0.22 \pm 0.07$ m w.e.a$^{-1}$) are in line with other geodetic measurements based on topographic maps and the SRTM DEM[52]. We measured a near balanced mass budget from 2000 to 2012 ($-0.03 \pm 0.02$ m w.e.a$^{-1}$), akin to other studies based on optical satellite data ($-0.07 \pm 0.05$ m w.e.a$^{-1}$)[16] and microwave data ($-0.04 \pm 0.02$ m w.e.a$^{-1}$)[53]. Our most contemporary (2012–2019) mass loss estimates provide evidence that glaciers of the Purogangri Ice Cap have again started losing ice ($-0.12 \pm 0.05$ m w.e.a$^{-1}$), which was also suggested by Liu et al.[54].

Glaciers in both the Western Nyainqentanglha mountain range and Poiqu region experienced gradually increasing ice mass loss rates throughout the observation period. Our data shows a substantial increase in ice loss rates in Western Nyainqentanglha glaciers over the past five decades, from $-0.26 \pm 0.13$ m w.e.a$^{-1}$ between 1968 and 2001 to $-0.43 \pm 0.12$ m w.e.a$^{-1}$ between 2001 and 2019. Independent studies covering comparable time periods[16,24] measured a similar acceleration in ice loss rates here. However, estimates based on SRTM and TerraSAR-X data ($-0.24 \pm 0.13$ m w.e.a$^{-1}$ and $-0.30 \pm 0.07$ m w.e.a$^{-1}$)[55,56] suggest less negative ice loss rates from 2000 to 2014. A source for this difference could include underestimation of the radar penetration in the SRTM dataset over clean ice glaciers in Western Nyainqentanglha.

The ice loss rate from glaciers in the Poiqu region also increased in recent decades (Supplementary Table 6). Our estimates of ice mass loss ($-0.30 \pm 0.10$ m w.e.a$^{-1}$ for 1974–2004 and $-0.42 \pm 0.11$ m w.e.a$^{-1}$ for 2004–2018) are similar to other geodetic observations covering similar time periods[16,26]. Our more temporally detailed measurements in the Langtang sub-region of the Poiqu region capture the consistent increases in mass loss rates here, from $-0.20 \pm 0.09$ m w.e.a$^{-1}$ (1964–1974) to $-0.59 \pm 0.14$ m w.e.a$^{-1}$ (2017–2019), which agree with more contemporary mass loss estimates of other studies covering longer time periods here[16,22,26].

The comparison of our results in the Poiqu/Langtang regions with temporally detailed studies of glacier mass loss at Mt. Everest suggests pervasive increases in the ice loss rate across the Central Himalaya in response to increased temperatures and decreased precipitation[57], which our climatic analyses also indicate over the nearby Poiqu basin. Glaciers located at Mt. Everest, south of the main orographic divide, have lost mass since at least 1970 ($-0.32 \pm 0.08$ m w.e.a$^{-1}$ for 1970–2007), with increased ice loss towards the end of the same period ($-0.79 \pm 0.52$ m w.e.a$^{-1}$ for 2002–2007)[33]. Another study also found consistent acceleration of glacier mass loss around Mt. Everest region between the 1960s ($-0.23 \pm 0.12$ m

w.e.a$^{-1}$) and the modern era ($-0.38 \pm 0.11$ m w.e.a$^{-1}$ from 2009 to 2018)[58].

We observed heterogeneous inter-decadal ice mass loss rates in Gurla Mandhata and the Muztagh Ata massif. Our measurements of ice loss since 2000 ($-0.15 \pm 0.09$ m w.e.a$^{-1}$ from 2000 to 2016) are consistent with other estimates of the contemporary ice loss rate ($-0.14 \pm 0.07$ m w.e.a$^{-1}$)[16], but our multi-temporal observations capture the transition from a balanced mass budget to mass loss over this period. We measured a near balanced or positive ($-0.02 \pm 0.08$ m w.e.a$^{-1}$) mass budget from 2011 to 2013, but moderate mass loss ($-0.24 \pm 0.17$ m w.e.a$^{-1}$) for 2018–2019, which is not evident in other less temporally resolved studies. Similarly, we measured an overall balanced mass budget ($-0.06 \pm 0.07$ m w.e.a$^{-1}$) over the last five decades (1967–2019) for glaciers in the Muztag Ata Massif, which is similar to other measurements of glacier mass balance in the region[21]. However, since 2013, ice mass loss has prevailed in Muztag Ata ($-0.12 \pm 0.11$ m w.e.a$^{-1}$ for 2013–2019) again suggesting the transition from balanced to negative glacier mass budgets here. Whilst broad scale geodetic studies have revealed the spatially heterogenous nature of glacier mass balance across HMA[16,17,40,59], our results emphasize the need for sub-decadal observations of glacier behavior to more accurately constrain the impact of climatic change on the cryosphere in HMA.

**Non-climatic factors affecting mass balance.** The additional consideration of non-climatic factors is, however, needed to fully explain glacier mass balance variability where only a weak link can be established between climate variables and glacier mass loss rates. For example, in Ak-Shirak, strong mass loss ($-0.54 \pm 0.10$ m w.e.a$^{-1}$) from 1980 to 2000 did not coincide with a marked increase in temperature or a decrease in precipitation in either the ERA5 Land or meteorological station data. Several glacier surges occurred during our 1964–1980 study period[60] in Ak-Shirak. Whilst these surge events were ongoing, surge-type glacier mass loss rates were slightly but insignificantly lower ($-0.27 \pm 0.16$ m w.e.a$^{-1}$) than non-surge-type glaciers ($-0.32 \pm 0.16$ m w.e.a$^{-1}$) in the region. However, following surge cessation the mass loss rate of the surge-type glaciers increased substantially ($-0.62 \pm 0.10$ m w.e.a$^{-1}$ from 1980 to 2002) and remained elevated above the mass loss rate of non-surge-type glaciers until the end of our time series ($-0.51 \pm 0.12$ versus $-0.41 \pm 0.12$ m w.e.a$^{-1}$ from 1980 to 2019). Such differences are likely due to the transfer of ice mass from high to low elevation during a surge event, where it would be more prone to melt, which occurred simultaneously on several glaciers in Ak-Shirak in the 1970s and 1980s. Less obvious differences in mass loss rates are apparent amongst the surge-type glaciers in Muztag Ata, where regional mass loss rates have been substantially lower than in Ak-Shirak. Surge events here have occurred irregularly over the past few decades[60,61] rather than in a temporally constrained manner. Over the full study period (1967–2019) in Muztag Ata the mass balance of surge-type glaciers ($+0.01 \pm 0.07$ m w.e.a$^{-1}$) was not substantially different from non-surge-type glaciers ($-0.07 \pm 0.07$ m w.e.a$^{-1}$) and less clear differences in mass loss rates were evident for surge-type and non-surge-type glaciers in different sub-periods (Supplementary Table 28).

Similarly, the exacerbation of glacier mass loss rates in response to proglacial lake expansion[26], which enhances mechanical calving and subaqueous melt, may have influenced the regional mass budget in Ak-Shirak and the Poiqu basin. The area of the moraine-dammed proglacial lake of Petrov Glacier increased by 240% (from $1.52 \pm 0.04$ km$^2$ to $6.16 \pm 0.06$ km$^2$) throughout our study period (1964–2019). The mass balance of Petrov Glacier, the largest in the Ak-Shirak region, was

consistently more negative than the regional average over each period (Supplementary Table 28), particularly between 1980 and 2002, when the glacial lake expanded by $2.08 \pm 0.05$ km$^2$ (140%). Glacial lake expansion has been, and continues to be, substantial across the central Himalaya and is likely to have exerted a strong influence on glacier mass loss rates in regions where large clusters of glacial lakes are found, such as the Poiqu basin[26,62].

**Implications for HMA glaciers.** Our study reveals the complex behavior of HMA glaciers and how climate variability drove these changes back to the 1960s. The use of declassified Corona KH-4 images (Fig. 5) from the 1960s, Hexagon KH-9 from the 1970s and contemporary, high resolution datasets such as TerraSAR-X and Pléiades provide temporally detailed observations of glacier mass change over six decades, which were formerly scarce across HMA. Recent mass loss increased in most regions including those characterized by neutral balance during the first decade of the 21st Century.

We compared geodetic mass balance estimates spanning more than half a century with ERA5 Land climate (Fig. 4) data and data from meteorological stations (Table 1, Supplementary Figs. 33–34) to link glacier mass change variability to changes in SumT and SolP at a sub-decadal level, which improves our understanding of the sensitivity of HMA glaciers to a changing climate. Both gridded and in-situ records of meteorological variability suggest SumT increases to be the main climatic driver of increased mass loss from glaciers in both humid and cold and dry climatic regions. More subtle variability in precipitation may be enhancing glacier mass changes in regions where glaciers are particularly sensitive to changes in accumulation, such as on the Tibetan Plateau.

Summer temperature anomalies of ERA5 Land gridded data and our meteorological station data display a significant correlation ($r^2$ ranges from 0.39 to 0.91 and $p$ ranges from 0.05 to 0.001) (Supplementary Figs. 18–20). However, the correlation found between meteorological station and ERA5 Land precipitation anomalies is much weaker. This emphasizes the need for improved and extended in-situ measurements at high elevation and careful validation of gridded precipitation data.

The spatial and temporal heterogeneity of mass loss from surge-type and lake-terminating glaciers in different parts of HMA emphasizes the need to look beyond climatic variables to understand variability in glacier behavior into the future. A more detailed understanding of the current prevalence of surge-type glaciers and likely future distribution of lake-terminating glaciers will help our understanding of glacier behavior in several regions across HMA. Similarly, the tighter constraint of the timing and magnitude of future glacier recession can only be achieved with observational data of improved spatial and temporal resolution. This study is a first attempt of that kind, and our mass change data can also be used to improve the calibration and/or validation of physically based modeling efforts.

## Methods
**Glacier outlines.** We used RGI V6.0 glacier outlines[63] as the baseline representation of glacier area in this study, which were modified manually using the orthorectified images of respective years. We manually adjusted clean ice and debris covered glacier tongues using surface slope and curvature, as well as shaded relief generated from the respective DEMs[64] as additional information. Subsequent adjustments were carried out for the other available time periods by taking advantage of the ortho images and DEMs.

**ERA5 Land gridded climate data.** We used Python 3.7 within Anaconda 3 and conda cdsapi package to download ERA5 Land data in netcdf format (https://cds.climate.copernicus.eu/#!/home). We downloaded HAR V2 monthly data in netcdf format from https://www.klima.tu-berlin.de/index.php?show=daten_har2. We extracted the variables '2 m temperature' in Kelvin and "total precipitation" in meter from ERA5 Land data. The purpose of HAR V2 data was to further validate

ERA5 Land data. For this data we extracted "total precipitation" in mm/hr, "frozen precipitation" in mm/hr and "column integrated absolute water vapor flux" in kg/m/s for Northern Tien Shan, Gurla Mandhata and Muztagh Ata regions; (more details in supplementary Note 1).

**DEM Generation, geodetic mass balance, and uncertainties.** Our geodetic glacier mass balance estimates are derived from a time series of DEMs generated from Corona KH-4 images from the 1960s and the early 1970s, Hexagon KH-9 images from the mid-1970s and more recent spaceborne imagery including Advanced Spaceborne Thermal Emission and Reflection Radiometer (ASTER) and Pléiades data (Supplementary Note 1 and Supplementary Table 1). To minimize errors due to geometry and distortion of the Corona KH-4 imagery (Supplementary Fig. 1) our workflow generates digital elevation models (DEMs) using image distortion models developed by Remote Sensing Software Graz. The original geometry of Hexagon KH-9 images was restored by using the information of the fiducial marks at the four corners of each scene and 1081 reseau crosses present in the raw images, and local adaptive filtering enhanced the contrast of the digitally-scanned imagery which led to improved image matching. We extended our time series to the present day using multi-platform, high-resolution optical stereo imagery and X-band Synthetic Aperture Radar (SAR) data.

**Corona KH4 & Hexagon KH9 DEM generation.** All Corona KH-4 DEMs were generated by using the Remote Sensing Software Package Graz (RSG), developed by Joanneum Research Graz. Instead of processing entire images we have processed a subset of the forward and backward strips. Initially, the panoramic camera orientation parameters were calculated by using GCPs (~30), which were collected from terrain corrected Landsat panchromatic (15 m) images and the SRTM DEM, using the integrated point measurement tools of RSG. A Corona-adopted image distortion model based on a modified collinearity equation[29], developed by RSG, was used to set-up an initial geometric model and optimised with the GCPs with an RMS of triangulation residuals of ≤~2.5 pixels. Further, the stereo pairs were coregistered using a matching based automated tie point measurement tool of RSG, which used the Foerstner operator[65] to detect suitable points. The forward image is mostly used to identify the suitable points[36]. Due to high parallax error (>1 pixel) a polynomial rectification registration was used before epipolar registration. To improve the quality of the final matching process we calculated disparity prediction in both directions using the geometric model and an external DEM (the SRTM DEM)[36]. Three levels of matching disparity predictions were performed in four pyramid-based levels with different search windows (5×11 pixels in first level and 5 × 5 pixels for subsequent levels) in both directions. The dimensions of the search windows were constrained iteratively. At the first level corresponding matching pixels were found within a large search window which was progressively tightened to eliminate erroneously assigned pixels. Finally, quality (0.6 pixels) and back-matching threshold (0.98 pixels) filters, estimated from the histograms from 3 and 4 bands of the disparity prediction images, were used to identify and remove the mismatch pixels[36].

Hexagon KH-9 DEMs were generated in Leica Photogrammetry Suite (LPS) by considering a frame camera model with a focal length of 305 mm and flying height of 170 km. Brown's physical model[66] was used to compensate for unknown lens and film distortions. In order to generate a quality DEM, it is necessary to reconstruct the original geometry of the Hexagon KH-9 films by evaluating their reseau grid overlaid on the photograph. The reseau crosses away from the center undergo certain distortions due to the long duration of storage of the films. The original image geometry was reconstructed by calculating the actual position of each reseau cross based on the distance between two crosses (1 cm) and the interior orientation was solved by bi-linear interpolation[20]. The co-ordinates of the corner fiducial points were calculated manually by considering that the image principal point is coincident with the central reseau point co-ordinate. Prior to the mosaicking of the subsets, a 51 × 51 pixels local adaptive filter and histogram equalization were used to enhance the contrast. The external orientation was modified by using GCPs (~40 GCPs) with an RMS of triangulation of ≤ ~1 pixel, collected from terrain corrected Landsat images as horizontal reference and SRTM as vertical reference, and automatically generated tie points.

**Pléiades, SPOT, GeoEye and ASTER DEM generation.** Terrain extraction from contemporary high-resolution stereo data was realized with the OrthoEngine module of PCI Geomatica by using the Rational Polynomial Coefficient (RPC) functions model. For planimetric adjustment, ~15 evenly distributed GCPs (RMSE ~≤ 1 pixels) were collected from terrain corrected Landsat images and the SRTM data. Tie points were generated automatically by PCI using the SRTM DEM and we eliminated tie points with residuals higher than half of the image pixel size resolution. After bundle block adjustment with first order RPC adjustment, the residuals of the GCPs in the stereo model were less than a pixel in both x and y directions. Finally, DEMs, with 1 m (Pléiades & GeoEye) and 5 m (SPOT-6/7) spatial resolutions were generated using the semi global matching (SGM) algorithm. PCI Geomatica OrthoEngine provides a score image corresponding to the quality of the stereo matching of each pixel. We have applied a threshold of 0.7 to exclude the less accurate elevation pixels from the DEM.

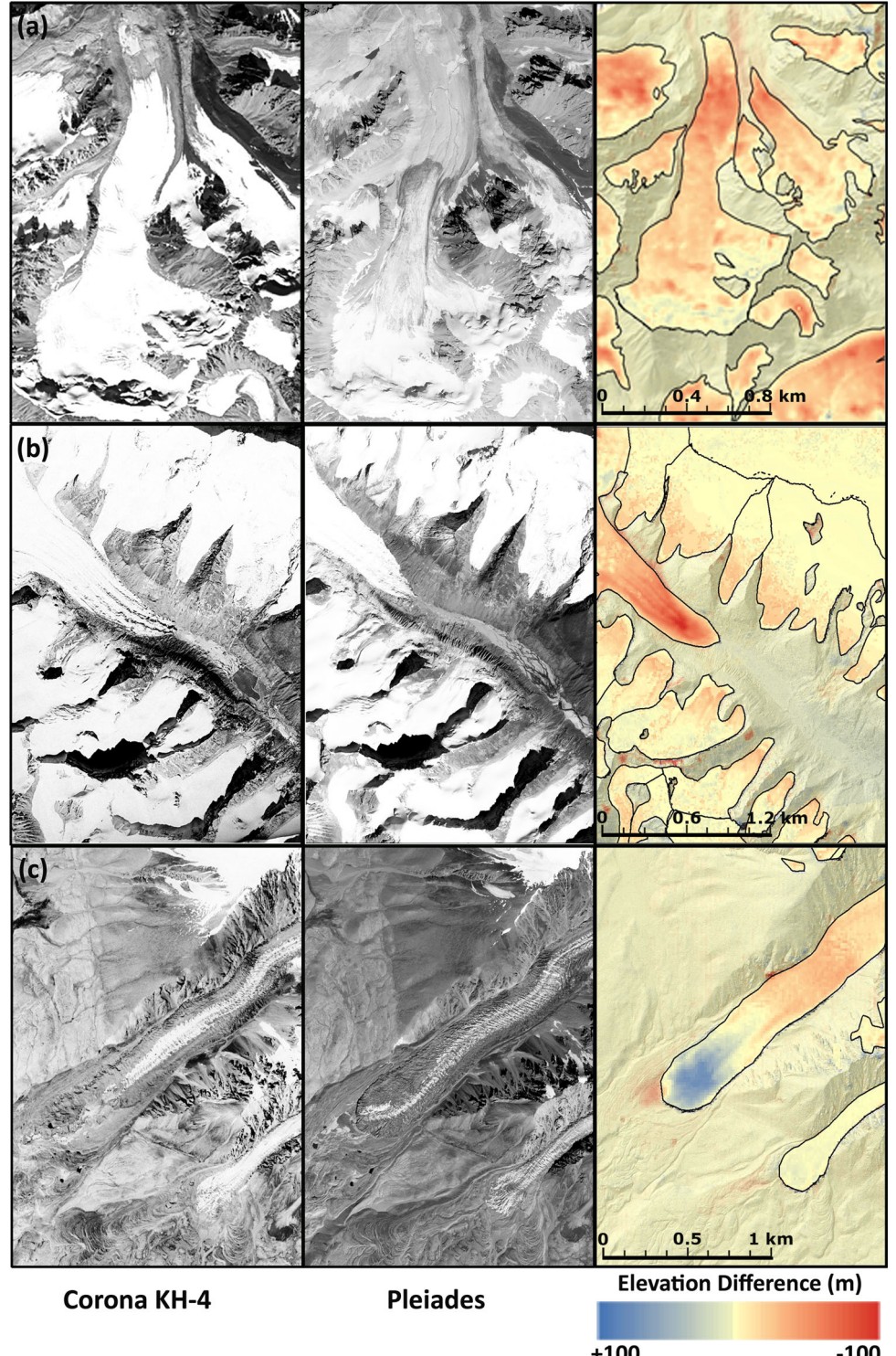

**Fig. 5 Visual comparison of declassified Corona KH-4 data with high resolution Pléiades data and corresponding elevation change image.** Subset images (Column wise) showing Corona KH-4 and Pléiades orthoimages (Pléiades, 27 Aug 2016, 29 Oct 2019, 5 Sep 2019 © CNES and Airbus DS (2016, 2019), all rights reserved) and the elevation difference between KH-4 and Pléiades DEMs for **a**. Northern Tien Shan, **b**. Western Nyainqentanglha, and **c**. Muztagh Ata Massif.

The open-source automated mass DEM production software Ames Stereo Pipeline (ASP)[64], developed by the National Aeronautics and Space Administration (NASA), was used to generate DEMs with a grid spacing of 30 m from the raw ASTER L1A stereo images. First, we extracted the nadir and backward-looking raw images and corresponding metadata files. Then we used RPCs to create an initial stereo pair and nadir and backward images were orthorectified afterward with 15 m spatial resolution and WGS84 datum. Further, a point cloud was generated using those orthorectified stereo pair of the images that overlap. Finally, 30 m spatial resolution gridded geocoded DEM was generated using the point cloud[67].

**TerraSAR-X DEM generation**. TanDEM-X DEMs were generated for Gurla Mandhata, Western Nyainqentanglha, and Purogangri Ice Cap. The Interferometric processing of the bistatic Co-registered Single-look Slant range Complex

(CoSSC) data provided by DLR was performed with the GAMMA SAR and interferometric processing software[68]. If more than one co-registered Single Slant Range Complex (CoSSC) data product was needed to cover the entire glacier area of interest, CoSSC data stripes were patched together to form one long data frame over the study area. From the CoSSC data, interferograms were formed employing $4 \times 4$ multi-looking. By employing the global TanDEM-X DEM[69] a differential interferogram was formed prior to phase unwrapping. The latter was filtered and unwrapped using a minimum cost flow (MCF) algorithm[70]. After the phase unwrapping the simulated phase of the global TanDEM-X DEM was added back to the differential interferogram resulting in a DEM with a time stamp of the respective CoSSC data acquisition. The final DEMs were geocoded and vertically adjusted to the global TanDEM-X DEM over stable terrain.

**Co-registration, outlier removal, and gap filling**. DEMs generated from different data sources were coregistered with the void filled 30 m SRTM DEM following the methods described by Nuth and Kääb[71]. Elevation dependent biases, present due to the tilt between two DEMs, were also estimated for each DEM using two-dimensional first order polynomial trend surfaces relative to the SRTM DEM[20]. Following DEM differencing between different epochs, all those pixels whose absolute elevation difference exceeded ±150 m were assumed to be obvious outliers and eliminated. Then the regional elevation difference was analyzed for each 100 m altitude bin and we consider those pixels whose absolute elevation difference values were within μ ± 3σ (μ and σ are the mean and standard deviation within the elevation bin)[47]. A few erroneous pixels were still found in Nyainqentanglha region where KH9 and TerraSAR-X data were involved. In order to remove those noisy pixels, instead of using regional elevation difference, we have analyzed individual glacier elevation differences for each 100 m altitude bin and removed those pixels whose absolute elevation difference values were outside μ ± 3σ. Considering the heterogeneity of the thickness change in glacierized terrain, outliers were removed by using an elevation dependent sigmoid function[50], assuming maximum thickness change in the ablation region and minimum thickness change in the accumulation region. The maximum allowable thickness change ($\Delta H_{MAXIMUM}$) for any pixel in a specific region was calculated using the Eq. (1)

$$\Delta H_{MAXIMUM} = \left[5 - 5\tanh\left\{2\pi - 5\left(\frac{E_{MAX} - E_{MIN}}{E_{GLAC}}\right)\right\}\right] \times STD_{GLAC} \quad (1)$$

Where, $E_{MAX}$ and $E_{MIN}$ are the maximum and minimum elevations for the whole region respectively, $E_{GLAC}$ is the glacier elevation of the pixel in meters and $STD_{GLAC}$ is the overall standard deviation of the glacier elevation differences. All values outside this range have been considered erroneous and removed. Data gaps were filled using a two-step approach. First, we used a $4 \times 4$ cell moving window to fill small data gaps with mean elevation change data from neighboring cells[26]. Secondly, larger data gaps in the ablation and accumulation regions were filled using median hypsometric methods[72] by calculating the median value of the elevation differences in every elevation bin for surge type and non-surge type glaciers separately. For this study, the median hypsometric void filling method has been considered to be more accurate as the median is more robust than the mean to any remaining outliers in the DEM difference images after outlier removal (see Supplementary Note 4).

Finally, the volume changes were converted into mass changes by considering the average ice density of 850 ± 60 kg m⁻³. The elevation difference over glaciers and stable terrain off-glacier are shown in Supplementary Figs. 2–9 and Supplementary Figs. 10–17, respectively.

**Radar penetration**. DEMs generated from TanDEM-X data may contain biases related to the penetration of X-band radar waves into dry snow and ice during data acquisition[73]. We incorporated TanDEM-X data into our time series of observations over PIC, Western Nyainqentanglha and Gurla Mandhata study areas, so further processing was required to quantify and eliminate potential X-band penetration biases in these regions. We examined the elevation differences between TanDEM-X DEMs and DEMs generated from optical data at coincident time periods to derive appropriate corrections for the TanDEM-X DEMs. For Gurla Mandhata, we compared a Pléiades DEM (26-10-2013) to the TanDEM-X (28-10-2013) and found mean elevation differences of 0.68 ± 0.28 m and 0.24 ± 0.18 m for clean ice and debris covered ice, respectively (Supplementary Fig. 35). This level of X-band penetration varied our mass balance estimates by just 0.02 m w.e.a⁻¹ over the period 2013–2016.

Comparable optical satellite imagery was not available for a date so close to the acquisition of TanDEM-X (26-01-2012) data over PIC. We have used ASTER data from 09-11-2012 here to estimate elevation differences, under the assumption that the low solid precipitation the area received between the two acquisition dates (~13 mm according to ERA5 Land climate data) has not substantially impacted glacier surface elevation data given by the ASTER DEM. We find a mean elevation change between the ASTER and TanDEM-X DEMs over PIC of 1.56 ± 0.68 m (Supplement Fig. 32). This magnitude of X-band penetration varied our mass balance estimates for the region by 0.04 m w.e.a⁻¹ over the period 2012–2018.

For Western Nyainqentanglha, which is located in a more humid climate than PIC, we have considered the error associated with X-band penetration to be ±0.03 m a⁻¹. This is based on the C-band penetration estimates (±0.07 m a⁻¹) derived for

Western Nyainqentanglha Zhang and Zhang[56], which are normally of greater magnitude than X-band radar.

**Seasonality correction**. To ensure that estimates of ice mass loss are not influenced by contrasting mass balance measurement or satellite image acquisition dates, a seasonality correction should be considered when images are acquired at different times of the year. The glaciers in our study regions are both winter-accumulation-type and summer-accumulation-type, depending on the region. Typical values for accumulation during winter months are about +0.15 m w.e. per month for northern Hemisphere glaciers, which is close to available measurements of winter-accumulation type glaciers in more humid parts of HMA[47]. The rate of accumulation of summer-accumulation-type glaciers located in continental climates are typically lower. We considered the best possible seasonal fit for the utilized satellite data, so that the glacier conditions with respect to accumulation and ablation are similar for the different acquisitions. The acquisition date of most of the scenes for each region deviate by less than two months (Supplementary Table 1) so that we consider the impact of the different acquisition dates is within the uncertainty range. The longest time difference for subsequent data for PIC is between 26th January 2012 and 27th September 2018. However, we also consider the variation to be within the uncertainty as the ice cap is of summer-accumulation-type, the precipitation is overall very low (see Supplementary Note 2) and a comparison to a DEM derived from an ASTER scene of 9th November 2012 revealed minor differences which can also be attributed to X-band radar penetration (Supplementary Fig. 32). We considered a correction for the first period of the data for Ak-Shirak (27th November 1964 to 31st July 1973). Measurements at Glacier No. 364 located in Ak-Shirak between 2011 and 2014 revealed an average accumulation of 0.035 m w.e. per winter month[74]. We adjusted the overall mass balance value for the mentioned period by 0.10 m w.e. A further correction was considered for Muztagh Ata Massif for the periods 4th August 1973 to 17th November 2001 and 17th November 2001 to 10th September 2009. Annual accumulation inferred from an ice core is about 0.6 m w.e. (on average 0.05 m w.e. per month) for the period 1958–2002[75]. Considering that most of the precipitation occurs between December and May[76] we performed only a slight correction of 0.05 m w.e. for the first period and a correction of 0.03 m w.e. for the second period. This correction leads also to a better fit to in situ measurements at Muztagh Ata Massif Glacier[21].

**Mapping uncertainty**. Glacier termini were mapped in a semi-automated fashion as described by Bjørk et al.[77]. To calculate the length change, we considered a reference point away from the terminus of the glacier. To do this, we approximated a centre flow line and found its intersection with the glacier terminus. The reference point has been considered as $3 \times W$ away from this point of intersection, where W is the width of the glacier near the terminus. We then divided the glacier terminus into 15 m apart equally spaced segments, connected the ends of all these segments with the reference point and calculated the average distance of all those from the terminus. The distance of the terminus from the same reference point at different times was finally used to estimate the effective length change.

Glacier length change uncertainty was calculated by considering the spatial resolution of the orthorectified images associated with different time periods. The uncertainty associated with terminus retreat ($U_{LENGTH}$) was estimated as follows[78]

$$U_{LENGTH} = \sqrt{(S_1)^2 + (S_2)^2} + RE_{REG} \quad (2)$$

Where, $S_1$ and $S_2$ are the spatial resolution of the imagery for first and second epoch respectively and $RE_{REG}$ is the rectification uncertainties. Though all the images are orthorectified, still there were some misalignments due to different software used for raw image processing. We have considered the RMSE associated in co registration as rectification uncertainty.

**Area uncertainty**. A mapping inaccuracy of 8 m (2 pixels) and 7.6 m (1 pixel) was assumed for the outlines derived from Corona KH-4 and Hexagon-KH9 images, respectively[37]. Similarly, a mapping inaccuracy of 7.5 m (1/2 a pixel)[49] and 7.5 m (5 pixels) were assumed for medium (ASTER and Landsat ETM+ ~15 m spatial resolution) and high resolution SPOT6 dataset, respectively. For Pleiades data we assumed a similar order of inaccuracy (7.5 m or 15 pixels). The overall uncertainty for the area change was estimated by the law of error propagation.

**Mass balance uncertainty**. Overall mass balance uncertainty ($U_M$) was estimated as a combination of the uncertainties associated with (i) thickness change uncertainties ($U_{THICKNESS}$), the combination of off glacier elevation changes uncertainty ($\Delta U_{DEM}$) and the uncertainty of glacierized area ($\Delta U_{AREA}$), and (ii) uncertainty associated with volume to mass conversion. The elevation change uncertainty ($\Delta U_{DEM}$) was calculated according to (Gardelle et al.)[44].

$$\Delta U_{DEM} = \frac{1}{k} \sum_{i=1}^{k} \frac{\sigma_{\Delta h}(i)}{\sqrt{\frac{N(i) \times S_{DEM}}{2 \times SA}}} \quad (3)$$

Where, $\Delta U_{DEM}$ is the uncertainty of the measured elevation differences, $\sigma_{\Delta h}$ (i) is the standard deviation of the mean elevation change of the non-glacier terrain in the i'th altitude band, $N$ (i) is the total number of off glacier pixels in the i'th

altitude band, $S_{DEM}$ pixel resolution of the DEM difference image and SA is the spatial autocorrelation distance of the DEM difference image[44]. Autocorrelation may occur at different scales[79] and vary over different types of terrain[27]. Berthier et al.[80] considered an autocorrelation length of 500 m for DEMs with coarser resolution (40 m) for Alaskan glaciers. Here, we consider a mean value of 600 m (20 pixels)[26,58] as representative of autocorrelation over the scale and various types of terrain present in our study area. Further, to estimate the uncertainty due to change in glacier area, we weighted the off glacier elevation change uncertainty ($\Delta U_{DEM}$) by the glacier area hypsometry.

Total thickness change uncertainty ($U_{THICKNESS}$) was calculated as the quadratic sum of the elevation change and area change uncertainties. Finally, the overall mass balance uncertainty ($U_M$) was estimated using the following Eq. (4)[50]

$$U_M = \sqrt{\left(\frac{\Delta h}{t} \times \frac{\Delta \rho}{\rho_w}\right)^2 + \left(\frac{U_{THICKNESS}}{t} \times \frac{\rho_i}{\rho_w}\right)^2} \quad (4)$$

Where, $\Delta h$ glacier thickness change, $t$ is the observation period, $\rho_w$ is the density of water (999.972 kg m$^{-3}$), $\rho_i$ is the ice density (850 kg m$^{-3}$) and $\Delta \rho$ is the ice density uncertainty[81].

The density conversion factor is likely to be variable in space and time for surge type glaciers at different points of their surge cycle and also for short time period mass budget estimates[78]. A limited number of studies[14,82] have compared the impact of the different density scenarios on geodetic glacier mass balance estimates but do not suggest a significant impact on regional mass loss estimates.

In consideration of the variability in the density conversion factor over short time periods[82] we have increased the uncertainty range associated with the conversion factor where our observation period is ≤3 years. We have adopted an uncertainty ($\Delta \rho$) estimate of 150 kg m$^{-3}$ over the shortest time (1 year) period and an uncertainty ($\Delta \rho$) estimate of 100 kg m$^{-3}$ for time periods of 2–3 years[81]. Over longer time span we have considered $\Delta \rho = 60$ kg m$^{-3}$.

## Data availability

Utilized Corona KH-4, Hexagon KH-9, and ASTER data are available at www.earthexplorer.usgs.gov. Pléiades, SPOT, TerraSAR-X, and GeoEye are commercial. ERA5 land data is available from https://cds.climate.copernicus.eu/cdsapp#!/dataset/reanalysis-era5-land?tab=form. Weather station data is restricted. Geodetic mass balances for each glacier grids of surface elevation changes will be available from PANGAEA (https://www.pangaea.de) and also available for download at www.mountcryo.org and upon request from the authors. All other relevant non-restricted data are available from the authors upon request.

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

## Acknowledgements

This study was supported by the Strategic Priority Research Program of Chinese Academy of Sciences (XDA20100300) and the Swiss National Science Foundation (200021E_177652/1) and benefited from the research cooperation within the Dragon 4 program supported by ESA and NRSCC (4000121469/17/I-NB). We thank the University of St Andrews for supporting the Article Processing Charges. We are grateful to CNES/Airbus DS for the provision of the Pléiades satellite data within the ISIS program and the Pléiades Glacier Observatory facilitated by Etienne Berthier (LEGOS) and Delphine Fontannaz (CNES). BM acknowledges funding from the "Global Water Futures", National Sciences and Engineering Research Council of Canada and the Canada Research Chairs Program. NN received funding from the European Union's Horizon 2020 programme (No. 689443). TanDEM-X data were made available through German Aerospace Center proposals GLAC1054 and GLAC7208. We also acknowledge the use of Copernicus Climate Change service (C3S) ERA5-Land reanalysis data that contains modified C3S Information (2020). We are also thankful to NASA EARTHDATA (https://earthdata.nasa.gov/) for providing freely available ASTER Stereo scenes.

## Author contributions

T.B. and A.B. designed the study. A.B. generated all digital elevation models from optical data, N.N. from microwave data. A.B. analyzed all the mass balance data and prepared the supplement figures and tables. O.K. generated glacier outlines and produced Fig. 1. K.M. processed and analyzed all the ERA5 Land climate data and produced the graphs for all climate data. Y.W. and T.Y. provided the in-situ data from China and V.K. from Kazakhstan. K.M., B.M., O.K., and T.B. interpreted the climate data. T.B., A.B., and O.K. interpreted the results. T.B., A.B., and O.K. led the writing of the paper and all other co-authors contributed to refining the manuscript.

## Competing interests

The authors declare no competing interests.
