## [Peer Review File · Nature Communications]

REVIEWER COMMENTS

Reviewer #1 (Remarks to the Author):

In this study, authors have analysed DEMs and estimated glacier mass fluctuation since 1960s. They selected seven regions covering HMA. We can often find such studies on glacier mass fluctuations limited in the recent years, or target regions are limited. But, this study covers both long term (since 1960s) and wide regions (HMA), which have heterogeneous conditions of glacier fluctuation and climate. Then, this paper is the first paper that have revealed long term fluctuation of glaciers covering HMA. It was very impressive and timely work for glacier fluctuation in HMA.

But, I have some comments. I hope my comments will help to improve your manuscript.

<Main comments>

- 1) Page 5 Authors have wrote that they conducted multiple regression analyses for glacier mass changes and meteorological elements. Multiple regression analyses can obtain equation of multiple regression and can derives t values for each explanatory variables (in this case ST, SP), and absolute value of t indicates a stronger correlation to objective variable. You showed only r² and p values in the manuscript and didn't show the t value in the multiple regression analyses. (I estimated authors didn't make multiple regression analyses, but they did only make correlation between mass changes and each meteorological element.)

If you can get t value, you can evaluate that which meteorological factors have stronger relation with glacier mass changes. But, I think number of data set (analyzed period such as 1964-1973, 1973-1980, 1980-2002, 2002-2009, 2009-2017 => you can get 5 data set), is not enough to get significant results. This is big issue to get result of multiple regression analyses.

- 2) Authors have only insisted that glacier fluctuations have relation with meteorological data and the mass increase/decrease has caused by solid precipitation increase/temperature increase (example). That's all.

I think what is interesting here is that glacier fluctuation which cannot be explained by meteorological data. You wrote only in the supplement (L471) about glaciers in the Akshirak, 'One possible explanation for this period of strong mass loss could be heightened glacier surface lowering following several glacier surges before 1980.'

I recommend that the discussion should be wrote in the main text.

Further, you can analyze that mass changes of non-surge glaciers and of surge glacier. Are there any difference in mass changes between surged or non-surged? If you can get some difference, you can suggest that glacier fluctuation can not be predicted accurately without model taking into account surge dynamics at the end of manuscript. And you have discussed on only surge for cause of irregular mass changes. But, large glacial lake at the terminus of the Petrov glacier might be one of the reason of the large mass loss as your co- author, Tobias and Owen have published.

<Specific comments>

Main text

P5 L22, L24 ($r^2=.. p=...$) which region's value were used to calculate the r^2 and p ? All humid regions? Is it possible?

P5 L18 There is no ST data from 1960s-1970s in Fig. 2. You should refer Fig. S18, here.

All " 0 C " should be change to " °C "

Supplements

L345 ...in-situ measurements and the values of selected... => you compared not values, but anomaly.

All " 0 C " should be change to " °C "

Table S3 S10 S11 => What indicates the values written in thick font?

L465 "Petrov glaciers (-28.4 ± 0.4 m a⁻¹) showed the strongest retreat rates over our entire study period." => In this section authors wrote about surge, but, there is no description about moraine dammed-glacial lake at Petrov Glacier. The large retreat rate is obvious because of the calving. Authors should write the specific condition about Petrov Glacier (As I mentioned in the main comment)

P4 L4 'However, these glaciers lost mass since, at a rate of up to -0.24 ± 0.16 m w.e.a⁻¹.'

I think this sentence is incomplete.

Supplement, Supplementary information => should unify the name.

Reviewer #2 (Remarks to the Author):

The authors measured and analyzed the long term mass balance of several sub regions of HMA, by combining various remote sensing data sets. As stated by the authors, such a data set is quite interesting for revealing the long term trend and identifying correlations with climate data, but also can serve as a cal/val database for modeling of future glacier evolution. Thus, the presented results are certainly of high interest, especially, once the used data sets are publicly available (as proposed) and can be integrated to other studies.

The main paper has no serious flaws, however in the methods (particularly see supplement) there are some issues which are unclear and need certain revision. Main issues:

- error analysis (unclear and maybe erroneous)
- SAR penetration (use elevation weighting)
- ERA-5 precipitation data (use integrated water vapor)
- void filling (why median hypsometric? Glacier or region scales?)
- Correlation of AWS and ERA-5 data (correlation metric are missing)

see below for more details regarding these issues and more specific comments and questions.

Moreover, the supplement provides a huge amount of information, which is nice, but some of the text sections can be certainly condensed in Tables.

Comments:

P2

L25: Why is the potential to capture glacier changes in the accumulation regions higher? Not clear, higher resolution does not necessarily mean the images are less saturated.

P4

L21: Precipitation from reanalysis data has a high uncertainty, especially in mountain regions. More meaningful would be to consider the integrated water vapor content, which can be used to estimate trends in precipitation.

L27: Did you also compare In-situ and ERA5 precipitation? How is the correlation? (see also comment above)

P5

L15: 15mm per year?

L17-18: mass loss coincides with the increase in ST...

L 30 $30^0 = 1$

P6:

L13: please provide the name of the author, only the number is not reader friendly

L28/29: how can reanalysis data being use to do projections? Not meaningful

Methods:

P11:

L29: McNabb et al. 2019 showed that "mean" hypsometric filling generates more reasonable results as compared to "median" hypsometric filling. Why did you choose "median" filling. Did you apply the hypsometric void filling on glacier scales or region scales?

Figures:

Fig1: some of the year numbers are hard to read

Maybe you can change the colors of the mass changes. The black segments attract most attention, but actually that's NA. Maybe NA → white, and no mass change green or yellow, is the scale of the subsets the same? If not please provide scale bars

"for each pixel..." not clear what you mean? The subsets?

Delete interpretation of results, that's already done in the main text

Fig2: delete: "entire available period"

delete interpretation of results.

Why do the uncertainties of the mass balance show short term variations? Why are the ERA values not always centered on the mass balance periods?

Fig3: data of KH-4 images? Delete interpretation.

If you want highlight the suitability of the imagery, you should also provide the off glacier elevation changes in the right column

Table 1: Delete interpretation

Supplement:

I76: please provide the operation period of the KH-4 operations

I77: from 1996 onward?

L84: afterwards

L85: please replace "⁰" by "°" everywhere

L103: telescopes

I105ff. Please rephrase, it sounds like the satellites were launched twice. Be more precise

I111: multi spectral? NIR? Please correct

I117: MS-band ? Explain or introduce

I140: Away from which point at the terminus? Manually defined?

L159: why did you use 5x11 and not a squared window?

L2014: why do you add the simulated phase to the unwrapped diff. Interferogram? That's not meaningful, since the simulated interferogram is not unwrapped.

L205: how was the phase to height conversion factor estimated?

L214: by applying such a filter in regions with heterogeneous elevation change patterns, you might lose important data. (e.g. surging glaciers). Did you check for this issue? For more details see Dussaillant et al. 2019 (supplement)

L216: same as above. What about surge type glaciers. They might get filtered out.

L227: see comment above regarding median hypsometric filling

I228: did you do the hypsometric analysis per glacier or per region? Not clear

I240ff: why do you include TanDEM-X data for PIC and Gurla, since you have optical data available?

So not correction would be needed. Moreover, you show penetration differences for different elevations. Thus you should apply them elevation dependent, based on your revealed distribution, since the glacier area is also not spread equally across the different elevations.

I248: why did it not impact the glacier surface elevation?

L289: how did you estimate RE_reg

L300: how did you estimate SA. What are the values?

L301: how can you estimate the U_area from U_DEM? Not meaningful. Area and elevation change are two different variables and have different units!

306: there is a bug in U_M. The result will not have "mass units". the unit will be m/y since you divide density by density. Moreover you have to integrate over the glacier area! Please revise!

320: this section can be also condensed to on Table, which would be more reader friendly and shorten the very long supplement.

345ff. As stated above, ERA5 precipitation data has certain limitations in mountain regions. Therefore, you should correlate the AWS data with integrated water vapor content. Maybe, the correlations will be better

I360: most of the section can be summarized in a table, showing the general characteristics of the study sites.

L343: same as for I360

Fig S5: whats the reason for the noisy pattern in the upper row images?

Fig S7: 2018-2019: What the reason for the strip shaped pattern?

Fig S8: 2009-2013 and 2013-2019 What the reason for the strip shaped pattern?

Fig: S9-s15: please use a more narrow color bar e.g. -3 to 3 m/a. So it is easier to judge the quality and explain the reasons for the strong outliers in various maps

Fig: S16-S21: please provide any correlation statistics to judge the correlation.

Fig. S23ff: what means "overall", all periods of your analysis (long term) or overall including other studies? Not clear

Fig S32: see comment regarding fig. 2

Reply to comments on the manuscript

'Spaceborne Observations Reveal Half a Century of Mass Loss from High Mountain Asia Glaciers'

Atanu Bhattacharya, Tobias Bolch, Kriti Mukherjee, Owen King, Brian Menounos, Vassiliy Kapitsa, Niklas Neckel, Wei Yang, Tandong Yao

REVIEWER COMMENTS

Reviewer #1 (Remarks to the Author):

Comment on the 'Spaceborne Observations Reveal Half a Century of Mass Loss from High Mountain Asia Glaciers' by Bhattacharya et al.

In this study, authors have analysed DEMs and estimated glacier mass fluctuation since 1960s. They selected seven regions covering HMA. We can often find such studies on glacier mass fluctuations limited in the recent years, or target regions are limited. But, this study covers both long term (since 1960s) and wide regions (HMA), which have heterogeneous conditions of glacier fluctuation and climate. Then, this paper is the first paper that have revealed long term fluctuation of glaciers covering HMA. It was very impressive and timely work for glacier fluctuation in HMA.

But, I have some comments. I hope my comments will help to improve your manuscript.

Reply: We would like to thank the reviewer for their positive and constructive comments which provided valuable insight, which undoubtedly helped to further improve the manuscript.

<Main comments>

1) Page 5 Authors have wrote that they conducted multiple regression analyses for glacier mass changes and meteorological elements. Multiple regression analyses can obtain equation of multiple regression and can derives t values for each explanatory variables (in this case ST, SP), and absolute value of t indicates a stronger correlation to objective variable. You showed only r² and p values in the manuscript and didn't show the t value in the multiple regression analyses. (I estimated authors didn't make multiple regression analyses, but they did only make correlation between mass changes and each meteorological element.)

If you can get t value, you can evaluate that which meteorological factors have stronger relation with glacier mass changes. But, I think number of data set (analyzed period such as 1964-1973, 1973-1980, 1980-2002, 2002-2009, 2009-2017 => you can get 5 data set), is not enough to get significant results. This is big issue to get result of multiple regression analyses.

Reply: We thank the reviewer for picking up this issue. As the reviewer suggests above, the terminological mistake on our part portrayed our statistical analyses of glacier mass balance and meteorological elements incorrectly, which we have been careful to avoid in our resubmission. In our original submission we conducted several simple regression analyses to establish the correlation between mass changes and each meteorological element, which was incorrectly termed as multiple regression analysis.

*We agree with the reviewer that multiple regression analyses would be the preferable tool to identify the most influential meteorological factors associated with glacier mass balance in different study regions. However, as the reviewer suggests, such analyses require a much denser dataset than our time series of mass balance estimates could provide, despite the unprecedented temporal resolution we have been able to achieve. As a result, we could not change the general approach with which we analyse the correlation between different meteorological variables and glacier mass balance. However, we have been more careful to describe the methods we followed correctly in our resubmission. Specifically, we have modified the text as **"The correlation between glacier mass balance estimates and meteorological factors confirmed ST to be the primary control...."** (Page 6; **Line 4-5**).*

To strengthen our analyses, we have also compared The High Asia Refined Analysis (HAR, Version 2) precipitation and water vapour flux data (Wang et al. 2020), available from 2004 to 2018, with ERA5 Land and corresponding weather station data as suggested by the second reviewer. We don't find strong correlation between water vapour flux obtained from HAR V2 and the precipitation of ERA5 Land and weather station data (Supplement Table 20). However, precipitation data of HAR V2 and ERA5 Land showed good agreement with one another, which supports the reliability of ERA5 Land data.

2) Authors have only insisted that glacier fluctuations have relation with meteorological data and the mass increase/decrease has caused by solid precipitation increase/temperature increase (example). That's all. I think what is interesting here is that glacier fluctuation which cannot be explained by meteorological data. You wrote only in the supplement (L471) about glaciers in the Akshirak, 'One possible explanation for this period of strong mass loss could be heightened glacier surface lowering following several glacier surges before 1980.'

I recommend that the discussion should be wrote in the main text.

Reply: We would like to thank the reviewer for this recommendation. We have expanded our analyses of the mass balance of surge-type glaciers in Ak-Shirak and in other regions where surge-type glacier

behaviour has been documented and modified the text accordingly. As suggested, we have added following lines (P11, L21-L25) in discussion section.

"The spatial and temporal heterogeneity of mass loss from surge-type and lake-terminating glaciers in different parts of HMA emphasizes the need to look beyond climatic variables to understand variability in glacier behavior into the future. A more detailed understanding of the current prevalence of surge-type glaciers and likely future distribution of lake-terminating glaciers will help our understanding of glacier behavior in several regions across HMA".

3) Further, you can analyze that mass changes of non-surge glaciers and of surge glacier. Are there any difference in mass changes between surged or non-surged? If you can get some difference, you can suggest that glacier fluctuation cannot be predicted accurately without model taking into account surge dynamics at the end of manuscript. And you have discussed on only surge for cause of irregular mass changes. But, large glacial lake at the terminus of the Petrov glacier might be one of the reason of the large mass loss as your co- author, Tobias and Owen have published.

Reply: We would like to thank the reviewer for this recommendation. There are several surge-type glaciers in Ak Shirak and Muztagh Ata and we have now specifically compared the mass balance for surge-type and non-surge-type glaciers (Supplement table 25) in these two regions. Moreover, we provide information about the elevated mass loss of the lake-terminating Petrov Glacier. We have added text to the results and conclusions sections to discuss this glacier type variability in more detail. Both the newly added sections are mentioned below:

Page 10; L5-L25 and Page 11, L1-L2: *"The additional consideration of non-climatic factors is, however, needed to fully explain glacier mass balance variability where only a weak link can be established between climate variables and glacier mass loss rates. For example, in Ak-Shirak, strong mass loss ($-0.54 \pm 0.10 \text{ m w.e.a}^{-1}$) from 1980-2000 did not coincide with a marked increase in temperature or a decrease in precipitation in either the ERA5 Land or meteorological station data. Several glacier surges occurred during our 1964-1980 study period⁵⁷ in Ak-Shirak. Whilst these surge events were ongoing, surge-type glacier mass loss rates were slightly but insignificantly lower ($-0.25 \pm 0.16 \text{ m w.e.a}^{-1}$) than non-surge-type glaciers ($-0.30 \pm 0.16 \text{ m w.e.a}^{-1}$) in the region. However, following surge cessation the mass loss rate of the surge-type glaciers increased substantially ($-0.62 \pm 0.10 \text{ m w.e.a}^{-1}$ from 1980-2002) and remained elevated above the mass loss rate of non-surge-type glaciers until the end of our time series (-0.51 ± 0.12 versus $-0.41 \pm 0.12 \text{ m w.e.a}^{-1}$ from 1980-2019). Such differences are likely due to the transfer of ice mass from high to low elevation during a surge event, where it would be more prone to melt, which occurred in a synchronous manner on several glaciers in Ak-Shirak in the 1970s and 80s. Less obvious differences in mass loss rates are apparent amongst the surge-type glaciers in Muztagh Ata, where regional mass loss rates have been substantially lower than in Ak-Shirak. Surge*

events here have occurred irregularly here over the past few decades here^{57,58} rather than in a temporally constrained manner. Over the full study period (1967-2019) in Muztag Ata the mass balance of surge-type ($+0.01 \pm 0.07 \text{ m w.e.a}^{-1}$) was not substantially different from non-surge-type glaciers ($-0.07 \pm 0.07 \text{ m w.e.a}^{-1}$) and less clear differences in mass loss rates were evident for surge-type and non-surge-type glaciers in different sub-periods (Supplementary Table 28)".

"Similarly, the exacerbation of glacier mass loss rates in response to proglacial lake expansion²⁶, which enhances mechanical calving and subaqueous melt, may have influenced the regional mass budget in Ak-Shirak and the Poiqu basin. The area of the moraine dammed proglacial lake of Petrov glacier increased by 240% (from $1.52 \pm 0.04 \text{ km}^2$ to $6.16 \pm 0.06 \text{ km}^2$) throughout our study period (1964-2019). The mass balance of Petrov Glacier, the largest in the Ak-Shirak region, was consistently more negative than the regional average over each period (Supplementary Table 28), particularly between 1980-2002, when the glacial lake expanded by $2.08 \pm 0.05 \text{ km}^2$ (140%). Glacial lake expansion has been, and continues to be, substantial across the central Himalaya and is likely to have exerted a strong influence on glacier mass loss rates in regions where large clusters of glacial lakes are found, such as the Poiqu basin^{26,59}".

<Specific comments>

Main text

1) P5 L22, L24 (r2=.. p=...) which region's value were used to calculate the r2 and p? All humid regions? Is it possible?

Reply: We have added following lines in our modified manuscript to better describe the grouping of study regions depending on their meteorological characteristics (P6, L9-L19).

"We therefore grouped the remaining regions in two clusters based on their general climatic characteristics and estimated the correlation between these compiled samples of glacier mass balance and climatic variables. Using annual precipitation and temperature data from ERA5 Land, we found that the mainly monsoon influenced Himalayan regions Poiqu and Gurla Mandhata have a humid climate with summer being the main precipitation season and mean annual precipitations of 1710 mm a^{-1} and 5220 mm a^{-1} and mean annual temperatures (MAAT) between $-2.4 \text{ }^{\circ}\text{C}$ and $-7.9 \text{ }^{\circ}\text{C}$. We also included Western Nyainqentanglha in this grouping (860 mm/a and $-3.75 \text{ }^{\circ}\text{C}$) with summer also being the main precipitation season. Ak-Shirak, Purogangri Ice Cap and Muztagh Ata Massif are typified by a cold, dry climate due to lower annual rates of precipitation (415 mm.a^{-1} - 607 mm.a^{-1}) and low annual temperature (-8.8°C to -9.7°C)."

2) P5 L18 There is no ST data from 1960s-1970s in Fig. 2. You should refer Fig. S18, here. All " 0 C " should be change to "0 C "

Reply: We changed the figure number (Supplement Fig. 20 and supplement table 18, P5, L28) in the main manuscript and also changed 0 c to °C in main manuscript and in supplement (This was a formatting issue in the earlier manuscript).

Supplements

3) L345 ...in-situ measurements and the values of selected... => you compared not values, but anomaly.

Reply: We agree and modified the text as follows (L189-L194, supplement Page 7)

"We observed a strong correlation of summer temperature anomaly between gridded ERA5 Land and meteorological data for all the regions (Supplement figs. 18-20) but, precipitation anomalies for most of the regions, except for winter precipitation of Gurla Mandhata ($r^2 = 0.82$ and $p = 0.01$), showed very little agreement. However, such a comparison might be biased since some of the AWS could have been used in the observational data used in the ERA5 product that was used to produce ERA5 land."

4) All " 0 C " should be change to " °C "

Reply: We replaced " 0 C " to " °C "

5) Table S3 S10 S11 => What indicates the values written in thick font?

Reply: We have highlighted those values which showed positive mass balance or glacier advance. For clarification, in our revised manuscript, we have added this information in the table captions.

6) L465 "Petrov glaciers ($-28.4 \pm 0.4 \text{ m a}^{-1}$) showed the strongest retreat rates over our entire study period." => In this section authors wrote about surge, but, there is no description about moraine dammed-glacial lake at Petrov Glacier. The large retreat rate is obvious because of the calving. Authors should write the specific condition about Petrov Glacier (As I mentioned in the main comment)

Reply: We thank the reviewer for raising this important point. We have calculated the size of the moraine dammed-glacial lake at Petrov Glacier throughout our time series of data over Ak-Shirak and highlight its likely impact on glacier mass loss rates in the region in the main manuscript (**Please see our reply 2**). We have also added following one sentence in the supplement (L330-L332, Supplement P12)

"The accelerated mass loss of the Petrov glacier might be due to the constant expansion (240%) of moraine dammed glacier lake throughout our study period (1964-2019)."

7) P4 L4 'However, these glaciers lost mass since, at a rate of up to $-0.24 \pm 0.16 \text{ m w.e.a}^{-1}$.' I think this sentence is incomplete.

Reply: We have completed the sentence as (P4, L18-L20)

"Following 2013, however, rates of mass loss from these glaciers increased to $-0.20 \pm 0.11 \text{ m w.e.a}^{-1}$ (2013-2016) and $-0.26 \pm 0.16 \text{ m w.e.a}^{-1}$ in our most recent time period (2018-2019)."

8) Supplement, Supplementary information => should unify the name.

Reply: We have changed and unify all as "supplement"

Reviewer #2 (Remarks to the Author):

The authors measured and analyzed the long term mass balance of several sub regions of HMA, by combining various remote sensing data sets. As stated by the authors, such a data set is quite interesting for revealing the long term trend and identifying correlations with climate data, but also can serve as a cal/val database for modeling of future glacier evolution. Thus, the presented results are certainly of high interest, especially, once the used data sets are publicly available (as proposed) and can be integrated to other studies.

Reply: We would like to thank the reviewer for their positive comments about the scope of our study. We agree that our results will be of benefit to the wider glaciological and hydrological community in studying the evolution of the Himalayan Cryosphere.

**The main paper has no serious flaws, however in the methods (particularly see supplement) there are some issues which are unclear and need certain revision. Main issues:
- error analysis (unclear and maybe erroneous):**

Reply: We thank the reviewer for their constructive and very useful comments, which have helped to improve various aspects of the manuscript. According to the Nature Communications structure guidelines, we have moved the methods section and our comparison of regional glacier mass balance estimates to other published studies from the supplement to the main text. We have also reanalysed some of the data and tried to improve our analyses based on the recommendations made by the reviewer. The error associated with the DEM difference data, identified by the reviewer, has been reanalysed according to the approach described later. Despite the additional processing, our mass balance results have not changed significantly, which increases our confidence in our approach (Supplement table 5,7, & 8).

(i) Fig S5 (Comment 43): We thank the reviewer for highlighting some noise evident in our elevation change results over Nyainqentanghla. On closer inspection, we identified small areas of erroneous pixels where KH9 and TerraSAR-X data were involved in DEM differencing. In order to remove those noisy pixels, instead of using regional elevation difference, we have analysed individual glacier elevation difference data over 100 m altitude bins and remove those pixels whose absolute elevation difference values were outside $\mu \pm 3\sigma$ (Supplement figure S5). The maximum elevation change (dh) differs from previous analysis by 0.22 m, which corresponds to a change in the mass budget of $+0.02 \text{ m w.e. a}^{-1}$ from 1968-1976.

(ii) Fig S7 (comment 44): We found slight misalignment between the DEMs (and resulting dh data) of 2018 and 2019 when they are coregistered with the SRTM DEM. This bias has been removed by coregistering the 2018 DEM with respect to 2019 DEM.

(iii) Fig S8 (comment 45): The strip on the DEM difference image for the year 2009-2013 & 2013-2019 occurred due to the presence of data gap in the raw Pleiades image of 2013. In order to remove the error before outlier removal we have manually identified all those pixels in DEM within the data gap by using both their hillshade, orthorectified image and their difference image relative to SRTM and set them to no-data.

- SAR penetration (use elevation weighting):

Reply: Instead of adding a mean penetration value derived across the entire region, we have corrected for X-band penetration across 100 m elevation bins by adding the mean penetration value for that particular elevation bin. This approach takes account of the spatial heterogeneity of X-band penetration biases and likely therefore provides a more robust correction through the elevation range of glacierized areas. The new approach recommended by the reviewer has not significantly impacted upon our mass balance estimates. For example, over the Purogangri Ice Cap, the mass budget over the two time periods involving TanDEM-X data (2012-2018) did not change significantly (from $-0.10 \pm 0.07 \text{ m w.e. a}^{-1}$ to $-0.12 \pm 0.05 \text{ m w.e. a}^{-1}$) due to this reprocessing.

- ERA-5 precipitation data (use integrated water vapor):

Reply: We have used the High Asia Refined Analysis (HAR, Version 2, 10 m resolution) water vapor flux and precipitation data, which is available from 2004 to 2018. We could have used the 'vertically integrated moisture divergence' from ERA5 data, but it is of much coarser resolution (30 km) than ERA5 Land data and higher resolution is very important to better represent the climate at the glaciers. On the other hand, ERA5 Land data does not have any variable representing water vapor flux. **Please see our detailed reply to this comment in the specific comment section (comment 2).**

- void filling (why median hypsometric? Glacier or region scales?):

Reply: To examine the sensitivity of our mass balance estimates to different void filling approaches, we examined the impact of glacier by glacier versus regional hypsometric void filling on our time series of elevation change data over Purogangri Ice Cap. Between 2000-2012, this region displayed an almost balanced glacier mass budget, thus the elevation change data over this period may be sensitive to the impact of inappropriate void filling techniques. In regions where surge type glaciers are present, we have applied a median hypsometric void filling approach separately for surge type and non-surge type glaciers. We have now better discussed and provided proper justification in support of our use of median hypsometric gap filling (please refer to reply of comment 9).

-Correlation of AWS and ERA-5 data (correlation metric are missing):
see below for more details regarding these issues and more specific comments and questions.
Moreover, the supplement provides a huge amount of information, which is nice, but some of the text sections can be certainly condensed in Tables:

Reply: In our revised manuscript we refer to the correlation metric which can be found in the supplement. Some parts of the supplement text have been removed and condensed in tables.

Comments:

1. P2 L25: Why is the potential to capture glacier changes in the accumulation regions higher? Not clear, higher resolution does not necessarily mean the images are less saturated.

Reply: We agree with the reviewer and modified the sentence (P2, L24-L29) in the main manuscript as.

“Declassified stereo imagery acquired by the Corona KH-4 satellite in the 1960s and early 1970s^{28,29} is of high (~2 m) spatial resolution and provides the opportunity to capture glacier change, even in the accumulation area, during a period when no other high resolution stereo data were available. Better spatial resolution is helpful to generate DEMs with a more accurate representation of glacier surfaces and, hence, more accurate mass balance estimates^{30,31}.”

2. P4 L21: Precipitation from reanalysis data has a high uncertainty, especially in mountain regions. More meaningful would be to consider the integrated water vapor content, which can be used to estimate trends in precipitation.

Reply: We appreciate the recommendation. We agree with the reviewer that the precipitation from reanalysis data has a high uncertainty over higher elevations. However, ERA5 Land data is the only high resolution (nearly 9 km) long term meteorological data available for these regions. Moreover, we think that the integrated vapor transport (flux), rather integrated vapor content, would reveal a much better correlation. Therefore, we have focussed our analyses in this direction.

P5, L21-L28: In our revised manuscript we have added vertically integrated water vapour flux data. We have compared higher resolution (10 km) High Asia Refined Analysis (HAR, Version 2) precipitation and water vapour data (Wang et al. 2020), available from 2004 to 2018, with ERA5 Land and corresponding weather station data for three selected study regions, Northern Tien Shan, Gurla Mandhata and Muztag Ata regions (Supplementary table 19). We see no or only a weak correlation between HAR V2 water vapour flux and the precipitation of ERA5 Land ($r^2 = 0.01-0.58$ and $p = 0.1-0.02$) and weather station data ($r^2 = 0.01-0.44$ and $p = 0.2-0.5$). However, precipitation data (winter, annual and solid) of HAR V2 and ERA5 Land showed good agreement with one another for these three-study regions ($r^2 = 0.60-0.86$ and $p = 0.01-0.004$ for Northern Tien Shan, $r^2 = 0.63-0.84$ and $p = 0.01-0.02$ for Gurla Mandhata and $r^2 = 0.81-0.89$ and $p = 0.01-0.005$ for Muztag Ata) which supports the reliability of ERA5 Land data (Supplement Table 19).

3. P4 L27: Did you also compare In-situ and ERA5 precipitation? How is the correlation? (see also comment above)

Reply: Yes, we have compared In-situ and ERA5 Land summer and winter precipitation (Supplement Fig. S18-S20). However, such a comparison might be biased since some of the AWS could have been used in the observational data used in the ERA5 product that was used to produce ERA5 land. Moreover, the correct measurement of precipitation in high mountain areas is notoriously difficult due to strong winds which make not only solid but also liquid precipitation difficult to measure. This is also evident from comparisons of gridded precipitation data which is based on station data such as Aphrodite. A study which adjusted high mountain precipitation based on glacio-hydrological modelling showed that Aphrodite significantly underestimates the precipitation while HAR1, the precursor of HAR2 (Wang et al. 2020), captured the amount of precipitation quite well (Wortman et al. 2018).

Our correlation values are provided in the Supplement Figures S18-S20. In general, we did not find a strong correlation for precipitation anomalies for most of the regions, except winter precipitation of Gurla Mandhata ($r^2 = 0.82$ and $p = 0.01$). However, summer temperature anomalies between the two data sets for all study regions provide strong significant correlation ($r^2 = 0.39-0.91$ and $p = 0.05-0.001$).

4. P5 L15: 15mm per year?

Reply: Solid precipitation increased during the period 2000-2012 by 42 mm a^{-1} or 15 % (P6, L24) compared to period 1975-2000.

5. P5 L17-18: ... mass loss coincides with the increase in ST...

Reply: The sentence has been modified (P6, L26-L29)

“mass losses have been coincident with increases in ST for two periods. ST increase between the 1960s and 1970s and after 2009 (Supplement Fig. 20 and supplement table 18) mirror increased rates of glacier mass loss”.

6. L 30 30⁰ =1

Reply: We are really sorry but we do not understand the comment properly.

7. P6 L13: please provide the name of the author, only the number is not reader friendly

Reply: Fully agree. We provide now the author name wherever possible and accordingly modified the text (for example: P8, L31).

8. P6 L28/29: how can reanalysis data being used to do projections? Not meaningful

Reply: This is a valid criticism. We have removed the sentence from the manuscript. We have now only provided the following sentence in our modified version (P11, L14-L20)

“Summer temperature anomalies of ERA5 Land gridded data and our meteorological station data display a significant correlation ($r^2 = 0.39-0.91$ and $p = 0.05-0.001$) (Supplement Figs. 18-20), however, there was no such correlation found between meteorological station and ERA5 Land precipitation anomalies. We also found a good agreement between the both the ERA5 land and HARv2 precipitation data. Such results emphasize the need for careful validation of gridded precipitation data prior to their use in glacier mass change modelling over larger regions”.

Methods:

9. P11: L29: McNabb et al. 2019 showed that “mean” hypsometric filling generates more reasonable results as compared to “median” hypsometric filling. Why did you choose “median” filling. Did you apply the hypsometric void filling on glacier scales or region scales?

Reply: This is a very important and relevant comment. Indeed the reviewer is correct that McNabb et al. (2019) mentioned that “mean” hypsometric filling generates more reasonable results compared to “median” hypsometric filling. However, it needs to be considered that McNabb et al. (2019) studied one time period and one region only, while we have seven regions and a number of different time periods. In our dataset we found that the distribution of elevation change data, after outlier removal, should be considered when choosing the method of filling.

To illustrate the impact of different approaches we assessed the sensitivity of elevation change values to the implementation of a regional mean hypsometric filling approach compared to regional median hypsometric filling, in two different cases, which are shown below. From the table it is evident that the void filling methods do not substantially affect values of elevation change if the elevation change

dataset is normally distributed (one example of the elevation change distribution of Ak-Shirak region for the year 1964-2019 is shown below). The second histogram, the elevation change distribution of Northern Tien Shan region for the year 1964-2020, shows a skewed distribution of elevation change. For this non-normally distributed data, different gap filling methods result in a slight but insignificant deviation in mass budget (-29.59 ± 6.7 m or -0.45 ± 0.09 m.w.e.a⁻¹ using mean hypsometry and -26.12 ± 6.7 m or -0.40 ± 0.09 m.w.e.a⁻¹ using median hypsometry). For this study area we have considered the regional median gap filling method to be more accurate as the median is more robust and less sensitive to any remaining outliers in the DEM difference images after outlier removal than the mean. For example, the median does not change substantially when we consider all elevation changes outside ± 100 m or ± 150 m as outliers, but the change in mean is more pronounced (Table 2). As we cannot be certain about the value of the obvious outlier, we have considered median to be more representative of the central tendency of the data.

In regard to the second point of the reviewer (regional versus glacier by glacier (local) derivation of gap filling), we have applied median hypsometric void filling at the regional scale over the majority of our study areas. However, in some study areas, such as Ak-Shirak, Purogangri and Muztagh-Ata Massif, where surge type glaciers are present, we have applied median hypsometric void filling separately for surge-type and non-surge type glaciers. This approach was required because the mass displacement involved in glacier surging is of a much greater magnitude than climate induced glacier thinning (or thickening) over the same time scale, thus surge-type glacier elevation changes would not be representative of the wider glacier population in the case of regional hypsometric void filling.

Where surge-type glaciers are not prevalent, we examined the impact of applying median gap filling on a regional or glacier by glacier (local) basis. We have computed the resulting mass balance of glaciers in the Purogangri Ice Cap (PIC) region over the period 2000-2012 using both regional and glacier by glacier (local) median hypsometry gap filling. We chose this location and time period because of the near-balanced glacier mass budget, which would be more sensitive to changes in our methodological approach. Gap filling using a regional median hypsometry yielded a near balanced mass budget for this time period (-0.03 ± 0.02 m w.e.a⁻¹, -0.31 ± 0.3 m.a⁻¹ elevation change). Median hypsometric gap filling on a glacier by glacier (local) basis yielded almost identical results (-0.02 ± 0.02 m w.e.a⁻¹ or -0.26 ± 0.3 m.a⁻¹ elevation change). This exercise confirms that our results are robust and the recalculation of all the data for all the periods and regions will not change the finding of our study.

Table 1: Comparison of Elevation difference in different time period using regional mean hypsometry and regional median hypsometry gap filling methods for Ak-Shirak region

Ak-Shirak Region (Elevation difference in meter)								
Gap Filling Method	1964-1973	1973-1980	1980-2002	2002-2009	2009-2015	2015-2017	2017-2019	1964-2019
Mean Hypsometry	-3.30 ± 1.92	-2.05 ± 1.70	-13.93 ± 3.00	-2.54 ± 1.04	-2.13 ± 1.05	-0.73 ± 0.84	-0.64 ± 0.60	-26.04 ± 5.31
Median Hypsometry	-3.21 ± 1.92	-2.15 ± 1.70	-13.98 ± 3.00	-2.59 ± 1.04	-2.12 ± 1.05	-0.76 ± 0.84	-0.68 ± 0.60	-26.13 ± 5.31

Figure 1: elevation change distribution of Ak-Shirak region for the year 1964-2019 and Northern Tien Shan region for the year 1964-2020

Table 2: Mean and median elevation change considering outlier outside ± 100 m or ± 150 m for Ak-Shirak and Northern Tien Shan region

Region	Ak-Shirak		Northern Tien Shan	
	Mean dh (m)	Median dh (m)	Mean dh (m)	Median dh (m)
Dh > 100 m	-23.76	-22.07	-26.90	-23.25
Dh > 150 m	-25.90	-22.58	-27.40	-23.39

Figures:

10. Fig1: some of the year numbers are hard to read

Maybe you can change the colors of the mass changes. The black segments attract most attention, but actually that's NA. Maybe NA → white, and no mass change green or yellow, is the scale of the subsets the same? If not please provide scale bars

“for each pixel...” not clear what you mean? The subsets?

Delete interpretation of results, that's already done in the main text

Reply: We adjusted several parts of Figure 1 according to the suggestions of the reviewer. We increased the font size of all labels and we adjusted the ‘no data’ colour for the segments of pie charts which represent the evolution of glacier mass balance in each region. We also changed the colour used to signify ‘no change’ in surface elevation $\pm 0.5 \text{ m a}^{-1}$ to a light yellow as the reviewer suggests, however we are not sure it improves the clarity with which the figure can be interpreted. We have therefore produced two versions of the Figure (light yellow versus white colour for ‘no change’ values) so either can be used, depending on the view of the reviewers and editor. The version with yellow colouring is shown below and the version with white colouring is included in the manuscript.

11. Fig2: delete: “entire available period”

Reply: We modified the figure (mentioned below) in our revised manuscript as per reviewer's suggestions

1. *We added recent time period data (2016-2020) for Northern Tien Shan*
2. *Removed the short-term variation of mass balance uncertainty by modifying the python script.*
3. *Deleted “entire available period” from figure caption*

12. delete interpretation of results.

Reply: Corrected.

13. Why do the uncertainties of the mass balance show short term variations? Why are the ERA values not always centered on the mass balance periods?

Reply: The short-term variation was due to a bug in our python script. We fixed the bug and accordingly modified all the figures in our revised manuscript (Supplement Figs S33-S34 & Fig. 3).

The ERA5 Land data is available from 1981 and therefore it has been centered at 1990-1991 for the period before 2000. For example, even if the period of mass balance is 1970-2000, it is centered at 1990-1991 and not at 1985-1986.

14. Fig3: data of KH-4 images? Delete interpretation.

Reply: As per reviewer’s suggestion we deleted the interpretation part from the figure caption (Figure 4). Therefore “data of KH-4 images” has been removed.

15. If you want highlight the suitability of the imagery, you should also provide the off glacier elevation changes in the right column

Reply: A good suggestion. We showed all off-glacier area of all regions in supplement figures but now show all terrain in the main text. Figure 4 has been modified accordingly.

16. Table 1: Delete interpretation

Reply: Corrected. We added the mass balance for recent time period (2016-2020) of Northern Tien Shan region.

Supplement:

17. I76: please provide the operation period of the KH-4 operations

Reply: We have modified the supplement in the following way (Supplement P4, L93-L94):

“Corona KH-4 satellites collected more than 860,000 images of the Earth’s surface between 1960 and 1972¹. The KH4 archives were declassified in 1995^{1,2}”.

18. I77: from 1996 onward?

Reply: Please refer to the reply to comment 17.

19. L84: afterwards

Reply: Modify as "**forward (FWD) and backward (AFT) directions**" (Supplement P4, L101-L102).

20. L85: please replace “^0” by “° ” everywhere

Reply: Replaced.

21. L103: telescopes

Reply: Telescope has been modified as telescopes in our revised manuscript (Supplement P5, L121).

22. l105ff. Please rephrase, it sounds like the satellites were launched twice. Be more precise

Reply: We have modified the text in our revised manuscript as (Supplement P5, L123-L126)

"SPOT-6 and SPOT-7, two optical remote sensing spacecraft of identical characteristics and capabilities of acquiring data of high resolution, were launched 2012 and 2014 respectively by Airbus Defense and Space (former name: EADS's Astrium Service) in order to continue sustainable wide-swath high resolution observation services¹¹".

23. l111: multi spectral? NIR? Please correct

Reply: Corrected (Supplement P5, L128)

24. l117: MS-band ? Explain or introduce

Reply: Text has been modified (Supplement P5, L135-L136)

"The nadir resolution of the panchromatic and four multi-spectral band (RGB and Near infrared (NIR)) is 0.7 m and 2.8 m, respectively".

25. l140: Away from which point at the terminus? Manually defined?

Reply: We consider the point where the approximate centre flow line intersects the glacier termini. The termini of the glaciers are delineated manually by visual interpretation.

As this method section has been shifted from supplement to main manuscript, we have modified the sentences as (P17, L8-L16)

"Glacier termini were mapped in a semi-automated fashion as described by Bjørk et al. (2012)⁷⁴. To calculate the length change, we considered a reference point away from the terminus of the glacier. To do this, we approximated a centre flow line and found its intersection with the glacier terminus. The reference point has been considered as $3 \times W$ away from this point of intersection, where W is the width of the glacier near the terminus. We then divided the glacier terminus into 15 m apart equally spaced segments, connected the ends of all these segments with the reference point and calculated the

average distance of all those from the terminus. . The distance of the terminus from the same reference point at different times was finally used to estimate the effective length change ".

26. L159: why did you use 5x11 and not a squared window?

Reply: This was a trial-and-error decision based on our KH-4 scenes. That means that it is generally case specific. The idea is that in the first level the corresponding pixels are found more safely and the size of the window is then reduced to avoid wrongly assigned pixels. This approach was tested manually for different sets of imagery and gave the best results.

We have added this information in our modified manuscript (P12, L32 and P13 L1-L3) as

“The dimensions of the search window were constrained iteratively. At the first level corresponding matching pixels were found within a large search window which was progressively tightened to eliminate erroneously assigned pixels”.

27. Interferogram? That’ s not meaningful, since the simulated interferogram is not unwrapped.

Reply: We respectively disagree with the reviewer on this point. For simulating a topography induced interferogram we used GAMMA's phase_sim_orb tool which simulates an unwrapped interferometric phase using DEM height and orbit state vector information.

28. L205: how was the phase to height conversion factor estimated?

Reply: Here we employed GAMMA's dh_map_orb tool which calculates the baseline for each point from DEM height, slant range, doppler centroid, and state vectors and estimates the derivative of the interferometric phase with respect to height. We employed the global TanDEM-X DEM as the reference elevation model which was also used for simulating the topography induced phase described above. The residual map then describes the difference between the global TanDEM-X DEM and the respective interferogram and hence needs to be added to the global TanDEM-X DEM to obtain the final DEM.

29. L214: by applying such a filter in regions with heterogeneous elevation change patterns, you might lose important data. (e.g. surging glaciers). Did you check for this issue? For more details see Dussailant et al. 2019 (supplement)

Reply: This is a fair and relevant comment. We followed Pieczonka & Bolch (2015) in our study, and outliers have been filtered by assuming a general non-linear trend of thickness change distributions for glaciers with negative budgets that typically have maximum lowering on their ablation zones and lowest lowering over the accumulation zones. However, this consideration might remove false pixels as an outlier for surge type glaciers with high frequencies of positive elevation change at comparatively low altitudes. Thus, heterogeneous glacier thickness changes with volume gains and volume losses in

ablation regions might be difficult to accommodate with a general threshold to the entire glaciated region in order to remove outliers. Therefore, Pieczonka & Bolch (2015) applied a filter considering the overall standard deviation of the glacier elevation differences weighted by an elevation dependent coefficient. With respect to the non-linear behavior of glacier thickness change the weighting coefficient has been determined using a sigmoid function which is supposed to be a good reflection of the elevation dependency. Additionally, we have removed the outliers for surge type and non-surge type glaciers separately using the methods described by Pieczonka & Bolch (2015).

30. L216: same as above. What about surge type glaciers? They might get filtered out.

Reply: Please refer to our reply for the previous comment (comment 29).

31. L227: see comment above regarding median hypsometric filling

Reply: Please refer to our reply for the previous comment (comment 9).

32. L228: did you do the hypsometric analysis per glacier or per region? Not clear

Reply: We are sorry for the unclear sentence in our manuscript.

Please also see our reply to comment 9. We have applied median hypsometric gap filling regionally. However, as mentioned above, we have considered surge type and non-surge type glaciers separately. We have modified the sentence as (P15, L13-L18)

" Secondly, larger data gaps in the ablation and accumulation regions were filled using median hypsometric methods⁶⁹ by calculating the median value of the elevation differences in every elevation bin for surge type and non-surge type glaciers separately. For this study the median hypsometric void filling method has been considered to be more accurate as the median is more robust than the mean to any remaining outliers in the DEM difference images after outlier removal (see supplement section 4)".

33. L240ff: why do you include TanDEM-X data for PIC and Gurla, since you have optical data available? So not correction would be needed. Moreover, you show penetration differences for different elevations. Thus you should apply them elevation dependent, based on your revealed distribution, since the glacier area is also not spread equally across the different elevations.

Reply: We chose to use TanDEM-X data only where no suitable, high-resolution optical data were available. We believe that TanDEM-X data yields a more robust, and spatially more complete, representation of the glacier accumulation area than low resolution ASTER optical data due to this sensor common saturation problem and resulting data voids in elevation change data. However, we still utilised ASTER data to derive X-band radar wave penetration estimates over glacier accumulation

areas covered by suitable ASTER scenes. We believe we have made the best use of both sensors with our approach as a result.

We agree with the reviewer that it is more robust to apply the derived X-band penetration estimates considering the magnitude of penetration at different elevations. Supplement Figure 35 shows the elevation dependant variability of elevation differences between different DEMs derived from optical imagery and TanDEM-X data over the Purogangri Ice Cap and Gurla Mandhata. We applied these different corrections in separate 100 m elevation bands to eliminate penetration biases from the TanDEM-X data.

34. L248: why did it not impact the glacier surface elevation?

Reply: We apologize for the typographical mistake in our manuscript. The precipitation was ~ 13 mm during this time period. The mistake has been rectified in modified manuscript (P16, L4).

35. L289: how did you estimate RE_reg

Reply: Though all the images are orthorectified, there may still be some misalignment due to the use of different software for raw image processing. We have estimated the RMSE (RE_{reg}) associated in co registration between two images which are used for length estimation (Hall et al., 2003).

36. L300: how did you estimate SA. What are the values?

Reply: Autocorrelation may occur at different scales (Dehecq et al., 2020) and vary over different types of terrain (Rolstad et al., 2009). Berthier et al. (2010) considered an autocorrelation length of 500 meter for DEMs with coarser resolution (40 m) for Alaskan glaciers. Here, we consider a mean value of 600 m (20 pixels) (Bolch et al., 2011; King et al. 2019) as representative of autocorrelation over the scale and various types of terrain present in our study area. We have added these sentences in our modified manuscript (P18, L11-L15)

“Autocorrelation may occur at different scales⁷⁶ and vary over different types of terrain²⁷. Berthier et al. (2010)⁷⁷ considered an autocorrelation length of 500 meter for DEMs with coarser resolution (40 m) for Alaskan glaciers. Here, we consider a mean value of 600 m (20 pixels)^{26,55} as representative of autocorrelation over the scale and various types of terrain present in our study area”.

37. L301: how can you estimate the U_area from U_DEM? Not meaningful. Area and elevation change are two different variables and have different units.

Reply: We agree with the reviewer’s comment. The statement was unclear in the manuscript. We have modified the statement in our revised manuscript (P18, L15-L19) to:

"Elevation change uncertainties will also vary for each individual glacier due to the different area distribution for each elevation bin. To estimate the elevation change uncertainty of glacier area spread across several elevation bands, we have calculated weighted averages of ΔU_{DEM} . ΔU_{DEM} for each individual elevation band is weighted by the glacier area hypsometry".

38. 306: there is a bug in U_M . The result will not have “mass units” . the unit will be m/y since you divide density by density. Moreover you have to integrate over the glacier area! Please revise!

Reply: It is true that the unit of U_M is meter/yr, instead of a mass unit. The dimension of mass balance, if expressed as a rate, is $[M T^{-1}]$, mass per unit time. When it is treated as a rate of change of mass per unit area, it is called specific mass balance and its dimension becomes $[M L^{-2} T^{-1}]$. When it is treated as a change of mass, it is called cumulative mass balance and its dimension becomes $[M]$ or $[M L^{-2}]$. When water-equivalent units are adopted, the dimension becomes $[L^3 T^{-1}]$, or $[L T^{-1}]$ for specific mass balance. In our study we have also used mass balance unit as water equivalent. To obtain the water equivalent, we have multiplied relative ice density (ice density/water density)

$$U_M = \sqrt{\left(\frac{\Delta h}{t} \times \frac{\Delta \rho}{\rho_w}\right)^2 + \left(\frac{U_{THICKNESS}}{t} \times \frac{\rho_i}{\rho_w}\right)^2}$$

This equation has been used in several studies (Huss 2013; Pieczonka & Bolch 2015, Goerlich et al. 2017 etc.). The unit $kg m^{-2}$ is usually replaced by the millimetre water equivalent (mm w.e.). This substitution is convenient because 1 kg of liquid water, of density $1000 kg m^{-3}$, has a thickness of exactly 1 mm when distributed uniformly over $1 m^2$. The unit $kg m^{-2}$ and mm w.e. are therefore numerically identical. More formally, the metre water equivalent (m w.e.) is an extension of the SI form.

39. 320: this section can be also condensed to on Table, which would be more reader friendly and shorten the very long supplement.

Reply: We prefer to keep this section in the supplement. However, we are agreed with the reviewer and as per the recommendation we removed the section 4 and presented in a tabular form in our revised version.

40. 345ff. As stated above, ERA5 precipitation data has certain limitations in mountain regions. Therefore, you should correlate the AWS data with integrated water vapor content. Maybe, the correlations will be better

Reply: Please refer to our reply for the previous comment (comment 2).

41. I360: most of the section can be summarized in a table, showing the general characteristics of the study sites.

Reply: Please refer to our reply for the previous comment (comment 39).

42. L343: same as for I360

Reply: Please refer to our reply for the previous comment (comment 39).

43. Fig S5: whats the reason for the noisy pattern in the upper row images?

Reply: We do not know the exact reasons, but reassessed the results and write the following:

P14, L31-33: "A few erroneous pixels were still found in Nyainqentanglha region where KH9 and TerraSAR-X data were involved. In order to remove those noisy pixels, instead of using regional elevation difference, we have analyzed individual glacier elevation differences for each 100 m altitude bin and removed those pixels whose absolute elevation difference values were outside $\mu \pm 3\sigma$ ".

44. Fig S7: 2018-2019: What the reason for the strip shaped pattern?

We have found a strip between the DEM difference image of 2018 and 2019. However, we have not found any strip when we subtract both 2018 and 2019 images with respect to SRTM DEM. The error might occur due to the erroneous coregistration. Therefore, we have reprocessed the 2018 DEM and coregistered it with 2019 DEM. The results are now fine.

45. Fig S8: 2009-2013 and 2013-2019 What the reason for the strip shaped pattern?

We found slight misalignment between the DEMs (and resulting dh data) of 2013 and 2019 when they are coregistered with the SRTM DEM. This bias has been removed by coregistering the 2013 DEM with respect to 2019 DEM.

46. Fig: S9-S15: please use a more narrow color bar e.g. -3 to 3 m/a. So it is easier to judge the quality and explain the reasons for the strong outliers in various maps.

Reply: The colour bar has been adjusted as recommended (Supplement figs S10-S17).

47. Fig: S16-S21: please provide any correlation statistics to judge the correlation.

Reply: The correlation statistics (r^2 and p value) have been added as recommended.

48. Fig. S23ff: what means "overall" , all periods of your analysis (long term) or overall including other studies? Not clear

Reply: Overall represents the entire observation period of our study. Accordingly, we have mentioned in the figure caption.

49. Fig S32: see comment regarding fig. 2

Reply: We have modified the figure as per recommendation.

Additional References:

1. Wang, X., Tolksdorf, V., Otto, M. & Scherer, D. WRF-based dynamical downscaling of ERA5 reanalysis data for High Mountain Asia: Towards a new version of the High Asia Refined Analysis. *Int. J. Climate*. 1-20 (2020), doi.: <https://doi.org/10.1002/joc.6686>.
2. Dehecq, A. Gardner, A.S., Alexandrov, O., McMichael, S., Hugonnet, R., Shean, D. & Marty, M. Automated Processing of Declassified KH-9 Hexagon Satellite Images for Global Elevation Change Analysis Since the 1970s. *Front. Earth Sci.* 8, 566802 (2020).
3. Rolstad, C., Haug, T., & Denby, B. Spatially integrated geodetic glacier mass balance and its uncertainty based on geostatistical analysis: application to the western Svartisen ice cap, Norway. *J. Glaciol.*, 55, 666–680 (2009).
4. Berthier, E., Schiefer, E., Clarke, G.K.C., Menounos, B. & Remy, F. Contribution of Alaskan glaciers to sea-level rise derived from satellite imagery. *Nat. Geosci.*, 3, 92-95 (2010).
5. Salerno, F. et al. Weak precipitation, warm winters and springs impact glaciers of south slopes of Mt. Everest (central Himalaya) in the last 2 decades (1994–2013). *Cryosphere*, 9, 1229–1247 (2015).
6. King, O. et al. Six decades of glacier mass changes around Mt. Everest are revealed by historical and contemporary images. *One Earth*, 3 (5), 608-620 (2020).
7. Mukherjee, K. et al. Surge-type glaciers in the Tien Shan (Central Asia). *Arct. Antarct. Alp. Res.* 49, 147–171 (2017).
8. Mölg, T., Maussion, F. & Scherer, D. Mid-latitude westerlies as a driver of glacier variability in monsoonal High Asia. *Nat. Clim. Change*. 4, 68–73 (2014).
9. Kang, S. Et al. Early onset of rainy season suppresses glacier melt: a case study on Zhadang glacier, Tibetan Plateau. *J. Glaciol.* 55 (192), 755–758 (2009).

REVIEWER COMMENTS

Reviewer #1 (Remarks to the Author):

Comment on the revised manuscript "Spaceborne Observations Reveal Half a Century of Mass Loss from High Mountain Asia Glaciers" written by Bhattacharya et al.

In the revised manuscript, I understood that you cannot use multiple regression analysis to decide primary control meteorological factors for glacier mass balance. Instead, you used P values to judge the strength of correlation between meteorological data and mass balance data.

And you also add section of 'Non-climatic factors affecting mass balance' according my comment. I think the section can be move to the supplement, if the number of words are too long.

As for the reliability of ERA5 data, I recommend to use weather station data (not ERA5) to analyze the relation between meteorological data and glacier mass fluctuations. Because reanalysis data has still have some error, in particular precipitation data.

Page 5 L23 – P6 L3

Comparison between HAR ver2 and ERA5 is not appropriate validation, because HAR v2 is a regional atmospheric data set generated by dynamical downscaling of global ERA5 reanalysis data using the WRF model, in other word, the correlation between HAR ver2 and ERA5 should be high. (as I wrote above) Why didn't you use station data? ERA5 data do not always represent reliable data as you indicated. I think you should make all discussion in 'Regional glacier response to climate change' using Supplement Fig. 33,34, not ERA5 data. But, you do not have to separate SP from WP. Annual precipitation is enough.

Page 6 L27-29

"ST increase between the 1960s and 1970s and after 2009 (Supplement Fig. 20 and supplement table 18) mirror increased rates of glacier mass loss. "

I cannot understand the description from Supplement Fig. 20 and supplement table 18. From Fig. 3, in Muztagh Ata Massif, median of STA in 2009-2011 was lowest since 2000, but mass loss increase constantly from 2000-2018.

Page 6 L20-22

"In cold, dry regions, glacier mass budgets were most strongly influenced by solid precipitation ($r^2= 0.79$, $p=0.0007$) and much less by summer or annual temperature ($r^2= 0.16-0.48$, $p= 0.1-0.17$). "

r^2 and p values of temperature have some ranges, but those of precipitation have only one values.

Page 6 L30

"(supplement table 18). "

Supplement table 18 indicates only show list of weather station.

Page 7 L15-18

"However, the location of Western Nyainqentanghla in a transition zone where both the Indian summer monsoon and mid-latitude westerlies exert an influence on local climate "

Authors cannot write the explanation without reference.

Supplement Fig. 33

I could not find the definition of anomaly. What is the standard value for those anomaly ?

Average of all period?

Only STA fluctuation in Gurla Mandhata looks low comparing to other region's data. The STA

might be WTA in Gurla Mandhata??

Reviewer #2 (Remarks to the Author):

The authors have addressed most comments adequately and there are only some minor issues left.

P2|26: It is still unclear why KH4 is able to capture elevations in the accumulation areas and KH9 not. The problem is not the spatial res. Of the imagery, it is the low contrast and saturation. Are there significant differences between KH4 and KH9? If so, please explain.

P3|25: remove comma after Poiqu

P3|27-29: too many "and"

P18|18: Did you do any weighting additional to Equation 3? The area per bin is already included ($N(i)$) in Equation 3? Unclear description. Please clarify

P18|25: Which density did you assume for the surge type glaciers? A surge is a dynamic process. Therefore it is most likely that all of the volume changes correspond to changes of the ice, which has a higher density. Also for post surge melt, the volume changes at the lower sections are caused by ice melt and not by changes of the firn pack. Maybe you can provide the estimates using both density scenarios. Would be interesting to see the impact. Moreover, according to Huss (2013), for short observation periods the uncertainty of the density is most likely much higher than 60 kg/m^3 . See e.g. Braun et al. 2019.

Fig. 1. The year numbers around the circles are still hard to read. Is the scale of the subsets the same?

Fig 4. The off glacier elevation changes are hard to see. Please reduce the transparency

Reply to comments on the manuscript

High Mountain Asian Glacier Response to Climate Revealed by Multi-temporal Satellite Observations since the 1960s

Atanu Bhattacharya, Tobias Bolch, Kriti Mukherjee, Owen King, Brian Menounos, Vassiliy Kapitsa, Niklas Neckel, Wei Yang, Tandong Yao

REVIEWER COMMENTS

Reviewer #1 (Remarks to the Author):

1. In the revised manuscript, I understood that you cannot use multiple regression analysis to decide primary control meteorological factors for glacier mass balance. Instead, you used P values to judge the strength of correlation between meteorological data and mass balance data.

And you also add section of 'Non-climatic factors affecting mass balance' according my comment. I think the section can be move to the supplement, if the number of words are too long.

As for the reliability of ERA5 data, I recommend to use weather station data (not ERA5) to analyze the relation between meteorological data and glacier mass fluctuations. Because reanalysis data still have some error, in particular precipitation data.

Reply: We would like to thank the reviewer for the positive and constructive comments.

The manuscript is well within the word limit and we would like to keep 'Non-climatic factors affecting mass balance' section in the main manuscript as it raises important points about glacial lake development and glacier surging and their impact on regional glacier mass balance.

We agree with the reviewer that reanalysis data will always somewhat differ from station data due to a number of factors (e.g. spatial resolution, dynamics not simulated by the mesoscale model used to produce the precipitation field in the reanalysis product) and these factors may be substantial in topographically complex terrain. We also note that measuring precipitation in high mountains is difficult and these high-elevation stations can be more biased than those at low elevation (e.g. Yang et al. 1999, Rasmussen et al. 2012). We stress, however, that our decision to use reanalysis data is to provide a way to consistently evaluate the response of temperature and precipitation across all glacierized regions of this study. The current observational network is too sparse to allow us to do that without reanalysis data.

Moreover, all the meteorological stations (except Tuyuksu) are installed at considerably lower altitudes compared to the mean glacier ELA (see supplementary table 18). Tuyuksu station is, like many others

in the former Soviet Union are Tretyakov type (see Fig. 1) which considerably underestimate precipitation (e.g. Yang et al. 1995).

Figure 1: Tuyuksu Meteorological Station in 2003 (Photo: T. Bolch)

Therefore, observed precipitation at meteorological stations might not be fully representative of higher altitudes. ERA5 Land data, the most recent high resolution (~9 km) climate reanalysis dataset, has good potential to capture the spatial variability of precipitation even at these higher altitudes. Moreover, the ERA5 Land reanalysis product is obtained by combining model data with observations considering the laws of physics to produce a globally consistent data set (Muñoz Sabater, J, 2019). To allow greater transferability of our results (across all of the regions of the present study and also for other glacierized regions for future work) we would prefer to maintain the comparison of mass change against ERA5 fields discussed in the paper.

In response to the reviewer's comments, we have analysed the correlation between glacier mass balance and meteorological factors using station data which we provide below for reference. These results indicate similar levels of sensitivity of glacier mass balance to particular meteorological variables as our previous analyses against ERA5 data. We have included some additional text about the in-situ observations, and show the results specific to weather station observations as a new table in the main manuscript and also added specific values in the related text. (Page 6, Line 12-23 and Table 2).

Page 6, Line 12-23: The generally weaker agreement of precipitation records (Table 2) likely reflects the relatively sparse network of weather stations and their typical location, aside from Tuyuksu (Northern Tien Shan), at lower elevations than glacierised terrain. Thus, in-situ measurements may not be fully representative of precipitation received by glaciers. In combination with the well documented difficulties associated with recording precipitation, high elevation meteorological stations may be more biased than those at lower elevations^{41,42}. On the other hand, the ERA5 Land reanalysis product

is obtained by combining model data with observations considering the laws of physics to produce a globally consistent data set ⁴³. To allow for the comparison of our mass balance data against a spatially and methodologically homogenous record of meteorological variables, we conduct further analyses considering the ERA5Land dataset primarily, but also provide a comparison our results to meteorological station records (Table 2).

Page 6, Line 26-28: A similar strong correlation between $\text{Sum}T_{\text{in-situ}}$ and glacier mass budgets exists for the measurements taken at the Tuyuksu meteorological station ($r^2= 0.92, p= 0.004$) (Table 2).

Page 7, Line 10-14: In cold, dry regions, glacier mass budgets were most strongly influenced by SolP. We find a particularly high correlation ($r^2= 0.79, p=0.0007$) between SolP in ERA5Land data and glacier mass balance in such regions (Table 2). However, the influence of temperature change could also be important here, as summer temperatures measured at weather stations showing substantial ($r^2= 0.60, p= 0.00005$) correlation with glacier mass balance estimates

Page 7, Line 29-34: In the more humid climatic regions (Poiqu, Gurla Mandhata and Western Nyainqentanglha), glacier mass budgets appear to be equally influenced by summer temperatures (Table 2) (ERA5 Land: $r^2 =0.71, p=0.003$; Weather station: $r^2=0.53 p=0.002$) and SolP (ERA5Land: $r^2=0.64$ and $p=0.0001$), annual (ERA5 Land: $r^2=0.64$ and $p=0.003$) and winter (ERA5 Land: $r^2=0.66, p=0.003$ and Weather station: $r^2=0.51, p=0.004$) precipitation

Page 12, Line 16-20: Both gridded and in-situ records of meteorological variability suggest $\text{Sum}T$ increases to be the main climatic driver of increased mass loss from glaciers in both humid and cold and dry climatic regions. More subtle variability in precipitation may be enhancing glacier mass changes in regions where glaciers are particularly sensitive to changes in accumulation, such as on the Tibetan Plateau.

Table 1. Correlation between different meteorological variables extracted from weather station observations, ERA5 Land and glacier mass balance estimates for different regions and climatic regimes. Bold numbers indicate strong correlation between glacier mass balance and meteorological variables.

Sl. No	Parameters	Northern Tien Shan (Tuyuksu station)		Humid climatic regions (GM, Poiqu, WN)		Humid climatic regions (GM, Poiqu)		Cold & Dry climatic regions (AK, PG, MA)	
		R^2	p	R^2	p	R^2	p	R^2	p
Weather Station Data									
1	Summer temp	0.92	0.004	0.53	0.002	0.77	0.004	0.60	0.00005
2	Summer Precipitation	0.67	0.06	0.52	0.15	0.57	0.23	0.39	0.64
3	Winter Precipitation	0.42	0.15	0.51	0.004	0.58	0.03	0.41	0.17

4	Annual Precipitation	0.59	0.06	0.38	0.15	0.62	0.27	0.44	0.83
ERA5 Land Reanalysis grided climate data									
1	Summer temp	0.97	0.009	0.71	0.003	0.69	0.001	0.16	0.10
2	Solid Precipitation	0.12	0.85	0.64	0.0001	0.56	0.001	0.79	0.0007
3	Summer Precipitation	0.01	0.86	0.66	0.3	0.60	0.2	0.19	0.15
4	Winter Precipitation	0.62	0.42	0.66	0.003	0.61	0.001	0.53	0.004
5	Annual Precipitation	0.19	0.78	0.64	0.003	0.56	0.001	0.62	0.02

* GM: Gurla Mandhata; WN: Western Nyainqentanghla; AK: Ak-Shirak; PG: Purogangri; MA: Muztagh Ata

2. Page 5 L23 – P6 L3

Comparison between HAR ver2 and ERA5 is not appropriate validation, because HAR v2 is a regional atmospheric data set generated by dynamical downscaling of global ERA5 reanalysis data using the WRF model, in other word, the correlation between HAR ver2 and ERA5 should be high.

Reply: We agree with the reviewer that this comparison is not perfect. We had used HAR V2 to address the following comment by one of the reviewers in the last review.

“Precipitation from reanalysis data has a high uncertainty, especially in mountain regions. More meaningful would be to consider the integrated water vapor content, which can be used to estimate trends in precipitation.”

We believe that it is the water vapor flux, and not the water vapor content, that would actually capture the trends of precipitation. Now we have removed comparison between HAR V2 and ERA5 from the main manuscript and add the following sentences in supplement text (P7, Line: 187-190):

Supplement text (P7, Line: 187-190): We found only a weak correlation (Supplement Table 19) between HAR V2 water vapour flux and the precipitation of ERA5 Land ($r^2 = 0.01$ to 0.58 and $p = 0.1$ to 0.02 for all three study regions) and weather station data ($r^2 = 0.01$ to 0.44 and $p = 0.2$ to 0.5 for all three study regions). Thus, we do not compare this dataset against our mass balance time series.

3. (as I wrote above) Why didn't you use station data? ERA5 data do not always represent reliable data as you indicated. I think you should make all discussion in 'Regional glacier response to climate change' using Supplement Fig. 33,34, not ERA5 data. But you do not have to separate SP from WP. Annual precipitation is enough.

Reply: Our response to comment 1 addresses the first part of the reviewer's question here. We have added a comparison of mass balance estimates to station data (Table 2) and we have directed the reader to this information in the main manuscript (Page 6, Line 10-21).

In regard to the separation of summer and winter precipitation, our intention was also to consider the seasonality of precipitation trends in our analysis, which is of specific importance in HMA as there are glaciers with different accumulation types (i.e. prevailing summer accumulation or winter accumulation). Hence, it is important to consider the seasonality and to distinguish between summer and winter precipitation. Our correlation analysis, based on ERA5 Land gridded data, captures the correlation between mass balance and seasonal changes in precipitation which provides key insight into climatic drivers of glacier mass change. For example, mass change in the humid regions of Gurla Mandhata, Poiqu and Western Nyainqentanghla (Table 2) are closely linked to both winter ($r^2= 0.66$, $p= 0.003$) and annual precipitation ($r^2= 0.64$, $p= 0.003$), however, glacier mass budget didn't show any dependency with summer precipitation ($r^2= 0.66$, $p= 0.3$). Similar results were obtained without considering Western Nyainqentanghla as a part of humid region, [winter precipitation ($r^2= 0.61$, $p= 0.001$), annual precipitation ($r^2= 0.56$, $p= 0.001$) and summer precipitation ($r^2= 0.60$, $p= 0.2$)]. Similarly, in cold and dry region, the mass balance was strongly linked with solid precipitation ($r^2= 0.79$, $p= 0.0007$) and also with winter ($r^2= 0.53$, $p= 0.004$) and annual ($r^2= 0.62$, $p= 0.02$) precipitation, however, no such correlation was found with summer precipitation ($r^2= 0.19$, $p= 0.15$)

4. Page 6 L27-29

“ST increase (author remark: at Muztagh Ata) between the 1960s and 1970s and after 2009 (Supplement Fig. 20 and supplement table 18) mirror increased rates of glacier mass loss.” I cannot understand the description from Supplement Fig. 20 and supplement table 18.

From Fig. 3, in Muztagh Ata Massif, median of STA in 2009-2011 was lowest since 2000, but mass loss increase constantly from 2000-2018.

Reply: The confusion is due to the wrong table number mentioned in the text. We are extremely sorry for the typographical error. The correct table number will be supplement table 17. In our revised manuscript we have modified the table number.

We agree with the reviewer's comment that the median summer temperature anomaly (STA) in Fig. 3 for Muztagh Ata massif did not show a significant difference, however, it is evident from the supplement table 17 that summer temperature (ST) increased $\sim 0.35^{\circ}\text{C}$ and $\sim 0.21^{\circ}\text{C}$ for the period after 2009 and 2013 respectively and is reflected by enhanced mass loss during those periods.

To avoid confusion, we have modified the text as (Page 7, Line 20-25)

Page 7, Line 20-25: At Muztagh Ata Massif, one of the coldest and driest regions in HMA, periods of ice mass loss have coincided with increases in SumT. SumT increase (~ 0.39 °C) and after 2009 (Supplement Fig. 20 and supplement table 17) accompanied increased rates of glacier mass loss. An increase in SumT (~ 0.21 °C) and decreased SolP (~ 10 mm a^{-1}) have likely driven mass loss after 2013 (supplement table 17).

5. Page 6 L20-22

“In cold, dry regions, glacier mass budgets were most strongly influenced by solid precipitation ($r^2=0.79$, $p=0.0007$) and much less by summer or annual temperature ($r^2=0.16-0.48$, $p=0.1-0.17$).” r^2 and p values of temperature have some ranges, but those of precipitation have only one values.

Reply: We are sorry that the information is not clear from the above sentences. We have changed the sentence in our modified manuscript as (Page 7, Line 10-14)

Page 7, Line 10-14: In cold, dry regions, glacier mass budgets were most strongly influenced by SolP. We find a particularly high correlation ($r^2=0.79$, $p=0.0007$) between SolP in ERA5Land data and glacier mass balance in such regions (Table 2). However, the influence of temperature change could also be important here, as summer temperatures measured at weather stations showing substantial ($r^2=0.60$, $p=0.00005$) correlation with glacier mass balance estimates.

6. Page 6 L30

“(supplement table 18).” Supplement table 18 indicates only show list of weather station.

Reply: The table number is incorrect. The correct table number (supplement table 17) has been mentioned in the modified manuscript.

7. Page 7 L15-18

“However, the location of Western Nyainqentanghla in a transition zone where both the Indian summer monsoon and mid-latitude westerlies exert an influence on local climate” Authors cannot write the explanation without reference.

Reply: We agree. We added the references Bolch et al. (2010) and Mölg et al. (2014) in the modified manuscript (P8, Line 7, Reference no. 44 & 45)

8. Supplement Fig. 33

I could not find the definition of anomaly. What is the standard value for those anomalies? Average of all period? Only STA fluctuation in Gurla Mandhata looks low comparing to other region's data. The STA might be WTA in Gurla Mandhata??

Reply: The anomaly is considered as deviation from the mean over the entire study period in each region which we have now mentioned in our revised manuscript (Page 5, Line 16-17)

It will be STA not WTA. However, we are thankful to the reviewer for pointing out the error in supplement Fig. 33. We have incorrectly calculated STA for Gurla Mandhata. In our revised manuscript we have rectified the mistake.

In our revised manuscript we have changed the acronyms we use for Summer temperature (SumT) and Solid precipitation (SolP) throughout.

Reviewer #2 (Remarks to the Author):

9. The authors have addressed most comments adequately and there are only some minor issues left.

Reply: We would like to thank the reviewer for their additional time devoted to reviewing our manuscript.

10. P2I26: It is still unclear why KH4 is able to capture elevations in the accumulation areas and KH9 not. The problem is not the spatial res. Of the imagery, it is the low contrast and saturation. Are there significant differences between kh4 and kh9? If so, please explain.

Reply: We agree with the reviewer that suitable contrast is needed to obtain good matching results. However, with higher resolution there is also higher probability to find matching points.

Therefore, we have modified the text in the manuscript as (Page 2, Line 25-31):

Page 2, Line 25-31: Declassified stereo imagery acquired by the Corona KH-4 satellite in the 1960s and early 1970s^{28,29} is of high (~2 m) spatial resolution, which increases the probability of image matching during DEM generation. Moreover, this data provides the opportunity to capture glacier elevation change during a period when no other high resolution stereo data were available^{30,31}.

11. P3I25: remove comma after Poiqu

Reply: Removed (Page 3, Line 31)

12. P3L27-29: too many “and”

Reply: Modified as (Page 3, Line 32 onwards)

Page 3, Line 32 onwards: Our results closely match those of previous studies for overlapping time periods (see Fig. 2 and section “Comparison with other mass balance estimates”) . However, our analyses extend these records back in time and towards the present day and also add intermediate periods.

13. P18L18: Did you do any weighting additional to Equation 3? The area per bin is already included (N(i)) in Equation 3? Unclear description. Please clarify

*Reply: We are sorry for the unclear sentence. Yes, to quantify the elevation change uncertainty of the glacier area distributed over the elevation bins, the **off-glacier** thickness change uncertainty (ΔU_{DEM}) is weighted by the hypsometry. The N(i) is defined as **off glacier** area in corresponding elevation bin, which was not mentioned previously in the manuscript. Therefore, the text has been modified as (Page 19, Line 25-30):*

Page 19, Line 25-30: Further, to estimate the uncertainty due to change in glacier area, we weighted the off-glacier elevation change uncertainty (ΔU_{DEM}) by the glacier area hypsometry.

9. P18L25: Which density did you assume for the surge type glaciers? A surge is a dynamic process. Therefore it is most likely that all of the volume changes correspond to changes of the ice, which has a higher density. Also for post surge melt, the volume changes at the lower sections are caused by ice melt and not by changes of the firn pack. Maybe you can provide the estimates using both density scenarios. Would be interesting to see the impact.

We assumed the same density scenario for all glaciers (850 kg per m³). We agree with the reviewer that the choice of a conversion factor throughout the different stages of a glaciers surge cycle is a complex issue. The surge of a glacier will redistribute a large volume of dense ice to a lower elevation where heightened ablation will then occur, which suggests a higher conversion factor may be more appropriate during the active- and post-surge phase. However, it must also be considered that shortly thereafter mass build up in the receiving zone will be mostly of firn and ice of a lower density. In addition, Jiskoot et al. (2001) and Reymond (1987) suggest increased 'porosity' due to crevassing during a surge, which can be extensive over both the reservoir and receiving zone of a surging glacier as a longitudinally extensive flow regime predominates. These two factors would both favour the use of a lower conversion factor, both during the quiescent and active phase of the surge.

A few previous studies have compared the impact of the different density scenarios on regional geodetic glacier mass balance estimates and did not find a significant impact when considering substantially different density scenarios. For example, Käab et al. (2012) calculated glacier mass balance over the Hindu Kush Karakoram Himalayan (HKKH) region using three different conversion factor scenarios; (i) overall density of 900 kg per m³ (ii) ice density of 900 kg per m³ and firn and snow density of 600 kg per m³ and (iii) the mean of the above two density scenarios. A more negative mass budget (-0.23 ± 0.05 m.w.e.a⁻¹) was reported by considering the overall density of 900 kg per m³. On the contrary, the least negative mass budget (-0.19 ± 0.04 m.w.e.a⁻¹) was found when considering different densities of ice and firn. The mass budget using the mean of the two different density

scenarios was well within the range of uncertainty ($-0.21 \pm 0.05 \text{ m.w.e.a}^{-1}$) of either alternative scenario. These results, which included mass balance estimates from surge-type glaciers, do not suggest huge differences would occur in regional mass loss rates when contrasting density scenarios are considered.

In addition, Braun et al. (2019) compared ice density scenarios of $850 \pm 60 \text{ kg per m}^3$ and $900 \pm 60 \text{ kg per m}^3$ in their study of geodetic glacier mass balance over Patagonian glaciers and concluded that $900 \pm 60 \text{ kg per m}^3$ is more representative of mass loss from glaciers losing ice through dynamic processes rather than alpine glaciers responding to climate variations. Again, the mass loss rate derived using both scenarios was not significantly different ($-0.58 \pm 0.07 \text{ m.w.e.a}^{-1}$ considering density $850 \pm 60 \text{ kg per m}^3$ and $-0.61 \pm 0.07 \text{ m.w.e.a}^{-1}$ considering density $900 \pm 60 \text{ kg per m}^3$). Clearly, it is difficult to disentangle these processes without much more temporally resolved datasets and is therefore well beyond the scope of the study. However, we have mentioned this issue in our revised manuscript as follows (Page 20, Line 8-12)

Page 20, Line 8-12: The density conversion factor is likely to be variable in space and time for surge type glaciers at different points of their surge cycle and also for short time period mass budget estimates⁷⁸. A limited number of studies^{14,82} have compared the impact of the different density scenarios on geodetic glacier mass balance estimates but do not suggest a significant impact on regional mass loss estimates.

10. Moreover, according to Huss (2013), for short observation periods the uncertainty of the density is most likely much higher than 60 kg/m^3 . See e.g. Braun et al. 2019.

Reply: We appreciate the reviewer highlighting the potentially large variability in the density conversion factor over short time periods and the need to consider an uncertainty range of greater magnitude as a result. In similar fashion to Braun et al. (2019) we have increased the uncertainty range associated with the conversion factor where our observation period is ≤ 3 years. We have carefully considered the results of Huss (2013) (Table 2 specifically) which suggest a departure of $\sim 150 \text{ kg m}^{-3}$ from the mean conversion factor (850 kg m^{-3}) over short (2 year) observation periods in three of the four scenarios considered in their study. We do not consider the results of 'Experiment III' in Huss (2013) (step change in glacier mass balance in combination with synchronous steepening/shallowing of the mass balance gradient) to be representative of any of the glaciers we have studied and therefore do not incorporate these results into our reconsidered uncertainty range associated with the conversion factor. Following Huss (2013) we have adopted an uncertainty estimate of 150 kg m^{-3} over the shortest time periods in our time series (1 year) and an uncertainty estimate of 100 kg m^{-3} for time periods of 2-3 years. We have not altered the uncertainty associated with the conversion factor for time periods

longer than 3 years (60 kg m^{-3}). These broader conversion factor uncertainty estimates do not cause large increases (± 0.01 to $\pm 0.02 \text{ m.w.e.}\alpha^{-1}$) in our overall mass balance uncertainty estimates, but we have added text to the methods section of the manuscript to better emphasise the issue raised by the review and to clearly describe our approach. We have added following text (Page 20, Line 13-17) to our manuscript to highlight the issue:

Page 20, Line 13-17: *In consideration of the variability in the density conversion factor over short time periods⁸² we have increased the uncertainty range associated with the conversion factor where our observation period is ≤ 3 years. We have adopted an uncertainty ($\Delta\rho$) estimate of 150 kg m^{-3} over the shortest time (1 year) period and an uncertainty ($\Delta\rho$) estimate of 100 kg m^{-3} for time periods of 2-3 years⁸¹. Over longer time span we have considered $\Delta\rho = 60 \text{ kg m}^{-3}$.*

11. Fig. 1. The year numbers around the circles are still hard to read. Is the scale of the subsets the same?

Reply: We have added scale bars to each regional subset and attempted to adjust the number around each pie chart to make them more visible.

17. Fig 4. The off-glacier elevation changes are hard to see. Please reduce the transparency

Reply: Modified accordingly.

References cited in the response:

- Bolch, T. et al. A glacier inventory for the western Nyainqentanglha Range and the Nam Co Basin, Tibet, and glacier changes 1976–2009. *Cryosphere*, **4**, 419-433(2010).
- Braun et al. Constraining glacier elevation and mass changes in South America. *Nst. Clim. Change.*, **9**, 130-136 (2019).
- Huss, M. Density assumptions for converting geodetic glacier volume change to mass change. *Cryosphere*. **7**, 877–887 (2013).
- Jiskoot, H., Pedersen, A.K., Murray, T. Multi-model photogrammetric analysis of the 1990s surge of Sortebræ, East Greenland. *J. Glaciol.* **47** (159), 677–687, (2001).
- Kääb, A., Berthier, E., Nuth, C., Gardelle, J. & Arnaud, Y. Contrasting patterns of early twenty-first-century glacier mass change in the Himalayas. *Nature* **488**, 495-498 (2012).
- Muñoz Sabater, J. ERA5-Land hourly data from 1981 to present. Copernicus Climate Change Service (C3S) Climate Data Store (CDS), doi: 10.24381/cds.e2161bac (2019).
- Mölg, T., Maussion, F., Scherer, D. Mid-latitude westerlies as a driver of glacier variability in monsoonal High Asia. *Nat. Clim. Change.*, **4**, 68-73 (2014).

- *Rasmussen, R. et al. How well are we measuring snow: The NOAA/FAA/NCAR winter precipitation test bed. Bull. Amer. Meteor. Soc. 93, 811–829 (2012).*
- *Raymond, C. F. How do glaciers surge? A review. J. Geophys. Res., 92 (B9), 9121–9134 (1987).*
- *Yang, D. et al. Accuracy of Tretyakov precipitation gauge: Result of WMO intercomparison. Hydrol. Process. 9, 877–895 (1995).*
- *Yang, D. et al. Wind-induced precipitation undercatch of the Hellmann gauges. Nordic Hydrol., 30, 57–80 (1999).*

REVIEWERS' COMMENTS

Reviewer #2 (Remarks to the Author):

The authors have addressed all comments. Thus, I would say, the paper can be published.

Congratulations

Thorsten